# IMPROVING FLOW FIELD PREDICTION OF COMPLEX GEOMETRIES USING SIMPLE GEOMETRIES

## ABSTRACT

In this study, we address the challenge of computationally expensive simulations of complex geometries, which are crucial for modern engineering design processes. While neural network-based flow field predictions have been suggested, prior studies generally exclude complex geometries. Our objective is to enhance flow predictions around complex geometries, which may often be deconstructed into multiple single, simple bodies, by leveraging existing data on these simple geometry flow fields. Using a case study of tandem-airfoils, we introduce a method employing the directional integrated distance representation for multiple objects, a residual pre-training scheme based on the freestream condition as a physical prior, and a residual training scheme utilising smooth combinations of single airfoil flow fields, also capitalising on the freestream condition. To optimise memory usage during training in large domains and improve prediction performance, we decompose simulation domains into smaller sub-domains, each processed by a different network. Extensive experiments on four new tandem-airfoil datasets, comprising over 2000 fluid simulations, demonstrate that our proposed method and techniques effectively enhance tandem-airfoil prediction accuracy by up to 96%.

## 1 INTRODUCTION

In engineering design processes, simulating flow around geometries via computational fluid dynamics (CFD) is a crucial step in modelling components. The study of partial differential equations (PDE) involving intricate and multiple geometries holds significant importance in various scientific and engineering applications, such as high-lift aircraft wings and wind farm wake interactions (Rumsey & Ying, 2002; Deskos et al., 2020). Analytical solutions are elusive due to the complexity of these problems. Constructing complex geometries by combining simpler shapes, though feasible, escalates simulation domains and resolution (Spalart & Venkatakrishnan, 2016), posing a challenge in terms of time and computational costs.

Previous efforts to replace costly simulations with neural networks have often treated each component independently and focused on simpler geometries scenarios with more abundant data (Pfaff et al., 2021; Lino et al., 2021; Fortunato et al., 2022; Cao et al., 2023). However, industries like aerospace and marine already simulate simple geometries like airfoils and hydrofoils extensively, offering an opportunity to exploit this data to efficiently predict flow fields for complex geometries. This paper introduces a novel method of leveraging extensive datasets from simple-geometry cases for more time-efficient predictions of flow fields in complex-geometry scenarios. The study focuses on tandem-airfoil, or a sequential arrangement of airfoils, chosen for its widespread application in various engineering domains including compressor blades in turbomachinery (McGlumphy et al., 2007), unmanned aerial vehicles (Yin et al., 2021; Okulski & Ławryńczuk, 2022), hydrofoil systems for maritime vessels (Maram et al., 2021), and race car engineering (Azmi et al., 2017).

The proposed method leverages the freestream condition as a physics prior (Ferm, 1990) to facilitate residual pre-training and smooth-combining of single airfoil predictions. Residual training, based on smoothly combined single airfoil flow fields, is then employed to improve tandem-airfoil predictions. To enhance performance and reduce GPU memory requirements, the simulation domain was decomposed, and different networks are trained for each sub-domain, allowing focus on specific flow regions. The directional integrated distance (DID), initially designed for single objects, is adapted to handle multiple objects effectively, addressing the numerical complexity that arises with increasing

object count. This work explores a novel problem setting—the use of simple geometry flow fields to improve predictions for complex geometries—through four new tandem-airfoil datasets with over 2000 CFD simulations[1]. Our contributions include:

- First study to use flow fields of simple geometries to improve predictions of complex geometries through a curriculum learning framework, demonstrated in the context of tandem-airfoil configurations.

- Novel use of freestream conditions as a physics prior for residual pre-training and smooth-combining of predicted flow fields from simpler geometries, significantly easing the training process.

- Adaptation of DID for multiple object representation and its integration into residual training using combined flow fields, differentiating from previous works using low-resolution flow field simulations.

- Establishment of four tandem-airfoil flow field datasets under different conditions for further exploration in future work (to be shared once this work is published).

## 2 PRELIMINARIES AND RELATED WORK

### 2.1 PRELIMINARIES

**Graph Construction**   The CFD simulation mesh $M$ is represented as a graph $G = (V, E)$, where $V$ and $E$ are the sets of nodes and edges, respectively. The mesh nodes are directly represented as graph nodes $i \in V$, and the faces between them as bi-directional edges $(i, j), (j, i) \in E$.

**Geometry Representations**   This work utilises the shortest vector (SV) and DID methods first proposed by Loh et al. (2024) to encode geometries as input node features. The SV is the shortest vector between the node and the geometry, while each DID value represents the average distance between the node and the geometry in a particular angle segment, with a predefined maximum distance $d_{max}$. However, these approaches were exclusively demonstrated on simple, single-geometries.

The DID was numerically estimated through the process summarised in Alg. 1. While extending the theoretical DID definition to accommodate multiple geometries is conceptually straightforward, the numerical calculations become progressively complex with each additional object.

---

**Algorithm 1** DID calculation. Steps that gain complexity with additional objects are shown in red.

**Input:** nodes $V$; positions $[(x_i, y_i) : i \in V]$; boundary indices $bd = [k \in V : k$ is on the boundary of a geometry$]$; angle segments $\left[ \left( \theta_j, \theta'_j \right) : 0 \le j < J \right]$; maximum $d_{max}$

$\text{DID} \leftarrow [\,]$
**for** $j \in [0, \dots, J-1]$ **do**
    $\text{DID}_j \leftarrow [\,]$
    **for** $i \in V$ **do**
        $d \leftarrow [$ distance between $i$ and $k, \forall k \in [k \in bd : (\theta_j < \theta_{i,k} < \theta'_j)$ and $(k$ is unobstructed from $i)] \,]$
        $d \leftarrow \text{minimum}(d, d_{max})$
        $\text{DID}_\theta \leftarrow$ average values of $d$
        $w_\theta \leftarrow$ proportion of $\left( \theta_j, \theta'_j \right)$ where $(k$ is unobstructed from $i), \forall k \in [k \in bd : (\theta_j < \theta_{i,k} < \theta'_j)]$
        $\text{DID}_i \leftarrow w_\theta * \text{DID}_\theta + (1 - w_\theta) * d_{max}$
    **end for**
    append $\text{DID}_i$ to $\text{DID}_j$
**end for**
append $\text{DID}_j$ to DID
**Return:** DID

---

**Solid Bodies within a Flow**   Flow around solid bodies is an important problem of fluid mechanics, which, in scenarios like aircraft design, is often modelled by an infinite flow region (Wu, 1976). Hence, the effect of the solid bodies on the flow can be thought of as a localised force field that can accelerate/decelerate the flow, diverting the latter to move around the bodies. Further away from the bodies, the flow will converge towards the so-called farfield boundary condition, essentially an

---

[1]Datasets will be made available to the public once this work is published.

undisturbed freestream flow at infinity (Fan & Li, 2019). So, flow velocity can be decomposed into:

$$\boldsymbol{U} = \boldsymbol{U}_\infty + \boldsymbol{U}' \, ,$$

where the overall flow velocity, $\boldsymbol{U}$, is the vector sum of the freestream conditions, $\boldsymbol{U}_\infty$, and the deviation from freestream, $\boldsymbol{U}'$, induced by the presence of a body and unique to its specific shape and orientation. Note that this decomposition is akin to the conversion from inertial frame to body frame (Speziale, 1998) and is also applicable to potential flows (superpositioning of inviscid freestream and singularity elements) (Collicott et al., 2017) and turbulent flows (steady mean and transient fluctuations) (Speziale, 1998). With that in mind, predicting flow fields with prior knowledge of freestream flow conditions and geometry motivates the smooth-combining and residual pre-training methods in Secs 3.1 and 3.3, respectively.

## 2.2 RELATED WORK

Previous neural network (NN)-based CFD strategies in fluid simulation have adapted in various ways. Graph neural networks (GNNs) have emerged as a primary method, demonstrating superior performance to MLPs and CNNs (Ogoke et al., 2020; Bonnet et al., 2022; Pfaff et al., 2021). Chen et al. (2021) introduced utilising both node and edge features, which are more suitable for CFD simulations and exhibited improved performance over basic GU-Net with vanilla graph convolutions, such as that used by Bonnet et al. (2022). Architectures employing encoder-processor-decoder, with similar graph convolutions, were adopted by Sanchez-Gonzalez et al. (2020) and Pfaff et al. (2021). Further enhancements through multi-scalability were introduced by Fortunato et al. (2022), Lino et al. (2021), Cao et al. (2023), and Gladstone et al. (2024), with the latter three achieving faster information propagation with a GU-Net style processor. These have set the trend for NN architecture and convolution design in CFD prediction, but they placed little focus on developing training techniques that leverage existing data or physical priors.

On the other hand, physical priors have been utilised through varying means, but still typically focused on flows with simpler geometries. Raissi et al. (2019) integrated physical equations into the training loss of NNs, later applied to Graph convolutional networks (GCNs) by Würth et al. (2024). This method reduces reliance on training data, but were shown on scenarios that involved no internal geometries. Some incorporate the use of physics-based solvers directly into the model, such as Obiols-Sales et al. (2020) to bring the network output to convergence, or de Avila Belbute-Peres et al. (2020) to physically simulate flow on a coarser mesh. Additionally, the work of Pfaff et al. (2021) was coupled with physics-inspired features and loss terms by Libao et al. (2023). However, none of these have considered cases involving the interaction of flows from two objects in tandem. Finally, Kochkov et al. (2021) further enhanced the concept of using the coarse solution in a method they referred to as "learned-correction", which capitalises on cost-effective data as the basis for estimation. They used a CNN to correct the errors in very low-resolution simulation results. More recently, Loh et al. (2024) exploited residual learning, an approach which had been extensively applied to image super-resolution (Zhang et al., 2018; Yang et al., 2019), in GCNs to enhance the training process and guide the model to concentrate on more intricate aspects of the prediction. However, it still required coarse physical simulations. A more recent work by Mao et al. (2024) has addressed domain decomposition for grid-based simulations using fixed-size subdomains and CNN-based architectures, but these approaches do not directly address mesh-based CFD scenarios involving complex geometries.

Hence, previous methods focused on balancing cost-accuracy trade-offs, but with a dearth of methods for complex geometries. Available public datasets, such as those provided by Bartoldson et al. (2023), are relatively few and are not appropriate for studying flows around multiple shapes in tandem by leveraging simpler geometry data. Reducing data dependence for complex geometries by such a method remains an area unexplored.

## 3 PROPOSED METHOD

In this section, we explain the schemes introduced in this paper, aiming to optimise the use of readily available simple geometry data for predicting complex geometry data, which is traditionally resource-intensive. The overall framework is illustrated in Fig. 1. It can be summarised as:

1. A simple-geometry neural network is pre-trained to predict single-airfoil flow fields given the geometry representations and boundary conditions. Importantly, freestream conditions serve as estimate fields for residual training.

2. The trained simple-geometry neural network predicts the single-airfoil fields, and the predictions are smoothly combined to generate combined fields with double-airfoils.

3. The neural networks in the complex geometry model are initialised with the weights of the pre-trained simple-geometry neural network.

4. These complex-geometry networks are then residually trained to predict double-airfoil flow fields, utilising the previously combined fields as estimate fields.

Note that the complex geometry model may encompass more than one neural network. Further details regarding this and each procedure within the overarching framework will be provided in subsequent sections, offering a detailed explanation of the entire methodology.

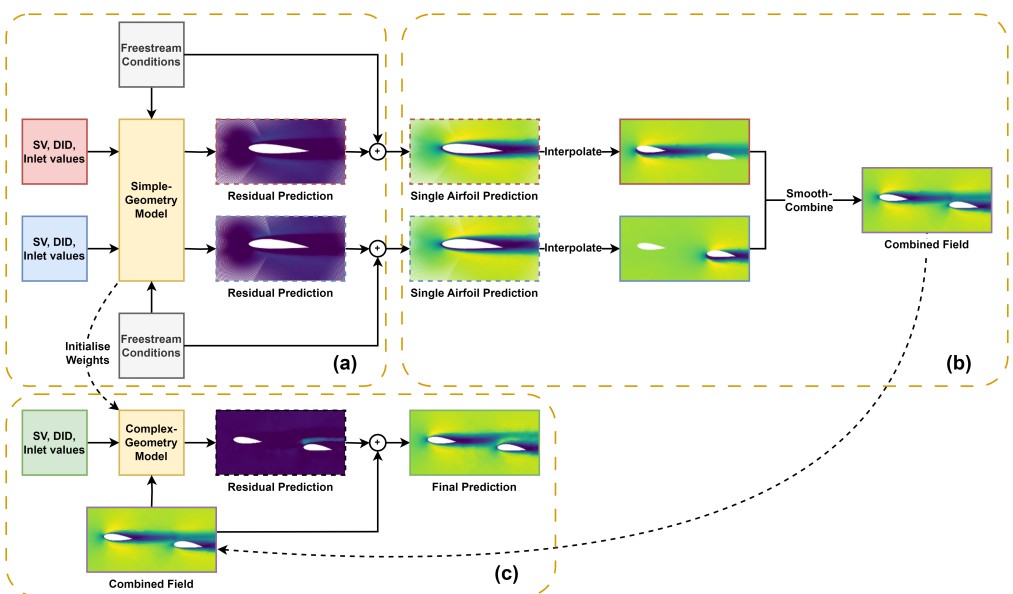

Figure 1: Overview of the proposed method, using (a) freestream-based residual pre-training, (b) smooth-combining, and (c) combined field-based residual training. Here, the tandem-airfoil model is portrayed as a single network for simplicity. A multi-NN as shown in Fig. 3 may be used instead.

## 3.1 SMOOTH-COMBINING

This section introduces the procedure for seamlessly combining multiple fields, with the objective to amalgamate the cost-effective fields of simple geometries to estimate the fields of more complex geometries (see Fig. 1(b)). Let $\boldsymbol{y}_1, \ldots, \boldsymbol{y}_L$ denote the $L$ fields to be combined. The combined field $\widetilde{\boldsymbol{y}}$ at node $i$ is then computed as:

$$\widetilde{\boldsymbol{y}}(i) = \boldsymbol{\gamma}_1(i) \cdot \boldsymbol{y}_1(i) + \cdots + \boldsymbol{\gamma}_L(i) \cdot \boldsymbol{y}_L(i) \ . \tag{1}$$

Here, $\boldsymbol{\gamma}_1, \ldots, \boldsymbol{\gamma}_L$ represent the weight of the respective original fields in the combined field. These will be assigned based on their absolute deviation from a reference field $\boldsymbol{y}_0$:

$$\boldsymbol{\gamma}_l(i) = \frac{|\boldsymbol{y}_0(i) - \boldsymbol{y}_l(i)|}{|\boldsymbol{y}_0(i) - \boldsymbol{y}_1(i)| + \cdots + |\boldsymbol{y}_0(i) - \boldsymbol{y}_L(i)|} \ . \tag{2}$$

At nodes $i$ where all fields do not deviate from the reference field, or $\boldsymbol{y}_1(i) = \cdots = \boldsymbol{y}_L(i) = \boldsymbol{y}_0(i)$, the weights can be set to $\boldsymbol{\gamma}_1(i), \ldots, \boldsymbol{\gamma}_L(i) = 1/L$. This results in the final combined field exactly matching the reference field at these nodes, i.e., $\widetilde{\boldsymbol{y}}(i) = \boldsymbol{y}_0(i)$.

**Deviation from Freestream**  When combining flow fields, we can assign weights to each field based on their deviation from the freestream. We will demonstrate using two single-airfoil fields to estimate a tandem-airfoil field (see Fig. 1(b)). Let $\boldsymbol{y}_1, \boldsymbol{y}_2 = \boldsymbol{U}_1, \boldsymbol{U}_2$ be two flow fields such as the x-velocity fields. Let $\boldsymbol{y}_0 = \boldsymbol{U}_\infty$ be the freestream flow field with no internal geometry. Then:

$$\widetilde{\boldsymbol{U}}(i) = \boldsymbol{\gamma}_1(i) \cdot \boldsymbol{U}_1(i) + \boldsymbol{\gamma}_2(i) \cdot \boldsymbol{U}_2(i) \ ,$$

$$\boldsymbol{\gamma}_l(i) = \frac{|\boldsymbol{U}_\infty(i) - \boldsymbol{U}_l(i)|}{|\boldsymbol{U}_\infty(i) - \boldsymbol{U}_1(i)| + |\boldsymbol{U}_\infty(i) - \boldsymbol{U}_2(i)|} = \frac{|\boldsymbol{U}_l'(i)|}{|\boldsymbol{U}_1'(i)| + |\boldsymbol{U}_2'(i)|} \ .$$

The approach is guided by the physical principles outlined in Sec. 2.1, which suggest that the influence of a solid body within a flow field can be conceptualised as deviation from the freestream, $\boldsymbol{U}'$. Hence, employing weights based on these deviations creates a combined field that effectively preserves the influences of both airfoils, resulting in a close estimate. The process is illustrated in Fig. 1(b). This novel concept not only ensures accuracy, but also offers computational efficiency due to its simplicity and the extra-low computational cost of the freestream conditions.

Figure 2 provides an example of the resulting weights, where the flow goes from left to right. Note that, where the flow fields are very similar but not equal to the freestream, the weights may still significantly favour the field that is most different from the freestream, such as in the blue area close to the front (left) airfoil. Refer to Appendix H for a further discussion.

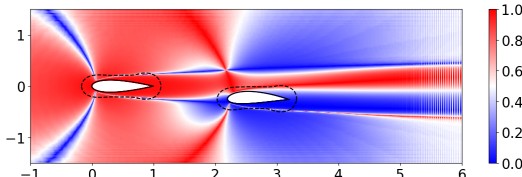

Figure 2: Weight values $\boldsymbol{\gamma}_1$ for the front airfoil.

### 3.2 DID Calculation for Multiple Objects

As mentioned, the SV and DID are incorporated into the neural network inputs as geometry representations. While both were utilised in previous applications, this marks the inaugural use of DID in a multiple-object scenario. Numerically calculating DID values using the original algorithm becomes progressively more complex and time-consuming with the addition of each object.

**Deviation from Maximum**  To address this challenge, we propose an alternative procedure that capitalises on the previously outlined smooth-combining scheme, using the deviation from the maximum value $d_{max}$ to weigh each field. This alternative method[2] is detailed in Alg. 2, providing an estimated DID representation of multiple geometries in a significantly reduced time-frame, ensuring computational efficiency of DID in a multi-object setting. Note that here, $\boldsymbol{y}_0 = d_{max}$, and $\boldsymbol{y}_1, \ldots, \boldsymbol{y}_L$ are the individual DID fields to be combined.

---

**Algorithm 2** DID estimation for $L$ number of objects.

**Input:** nodes $V$; positions $[(x_i, y_i) : i \in V]$; boundary indices $bd_l = [k \in V : k \text{ is on the boundary of geometry } l]$ $\forall$ geometries $l \in \{1, \ldots, L\}$; angles segments $\left[ \left( \theta_j, \theta_j' \right) : 0 \leq j < J \right]$; maximum value $d_{max}$

$\boldsymbol{y}_0 \leftarrow d_{max}$
**for** $l = 1$ **to** $L$ **do**
    $\boldsymbol{y}_l \leftarrow \text{DID}_l$ value calculated using Alg. 1 and boundary indices $bd_l$.
**end for**
$\text{DID}_{est} \leftarrow$ combined field $\widetilde{\boldsymbol{y}}$ calculated using Eqn. 1 and Eqn. 2, with $(\boldsymbol{y}_0, \ldots, \boldsymbol{y}_L)$
**Return:** $\text{DID}_{est}$

---

### 3.3 Residual Training

In this section, we will introduce two residual training schemes, one supporting the utilisation of pre-training, and the other capitalising on the smooth-combining techniques outlined in the previous section. Residual training, which is extensively used in image super-resolution, involves utilising an estimate to ease the learning. Let the network output be $\widehat{\boldsymbol{U}}$. Instead of directly predicting the flow field $\boldsymbol{U}_{gt}$, the model is trained to predict the residual field $\boldsymbol{U}_{gt} - \widetilde{\boldsymbol{U}}_{est}$ and minimise the loss function $\mathcal{L}\left( \boldsymbol{U}_{gt}, \widehat{\boldsymbol{U}} + \widetilde{\boldsymbol{U}}_{est} \right)$, where $\widetilde{\boldsymbol{U}}_{est}$ represents an estimated flow field. In CFD cases, this

---

[2]This method avoids performing Alg. 1 with multiple objects.

estimate often takes the form of a lower-resolution simulation result, which is more cost-effective to obtain than the corresponding ground-truth (Loh et al., 2024).

**Freestream Conditions** Rather than relying on a lower-resolution result, this paper suggests an innovative approach: using freestream conditions as an estimate for simple geometries, or setting $\widetilde{U}_{est} = U_{\infty}$. This concept, like the smooth-combining procedure, aligns with the physical principles discussed in Sec. 2.1 that assert that freestream conditions should serve as a reliable estimate for the majority of the field. In contrast to lower-resolution fields, freestream conditions do not need to be derived from any physics-based simulator, so its computational cost is minimal. The residually pre-trained network is used not only to predict the flow fields of single airfoils for smooth-combining, but also to initialise the weights of the network for predicting tandem-airfoil. This initialisation can improve the final prediction performance. Figure 1(a) illustrates the freestream-based residual pre-training.

**Combined Flow Fields** To estimate the flow field of tandem-airfoil, we propose employing combined flow fields, obtained through the smooth-combining procedure discussed in the previous section and illustrated in Fig. 1(b), or setting $\widetilde{U}_{est} = \widetilde{U}$. The combined flow field may still differ from the target tandem-airfoil flow field, but it provides a cost-effective estimate for improving the learning through residual training.

The combined field-based residual training process is visually presented in Fig. 1(c), where it can be seen as part of the consolidated pre-training, smooth-combining and residual training method. Both residual training procedures in the proposed method use different estimates, one based on the freestream conditions and the other on the combined flow fields, which are unlike previous methods based on low-resolution simulation results.

### 3.4 MULTI-NN INFERENCE PROCEDURE

Multi-NN inference involves training multiple NNs to predict extensive CFD domains with multiple geometries, like tandem-airfoil. While it is conceivable for the complex geometry model shown in Fig. 1(c) to be handled by a single neural network, training the network to predict a multi-geometry case after being pre-trained on solely single-geometry instances would be particularly demanding. The primary purpose of the multi-NN is hence to ensure that each neural network exclusively predicts a field with at most one geometry, mitigating these challenges.

A comparison between a single NN and the multi-NN inference procedure is depicted in Fig. 3. The multi-NN process can be summarised:

1. The CFD domain is subdivided into front, back, upper, and lower flow fields as in Fig. 4. Inputs like the SV, DID or estimated fields are segmented accordingly.

2. The front flow field is predicted, with inlet values serving as an input feature for nodes along the inlet. The remaining nodes in the rest of the field will receive a zero-value here.

3. The predicted values within the overlap regions between the front and back sub-graphs are then utilised as the corresponding input features for predicting the back flow field.

4. The previous step is repeated, employing both the inlet values and appropriate overlap data from the front and back fields, to compute the upper and lower fields.

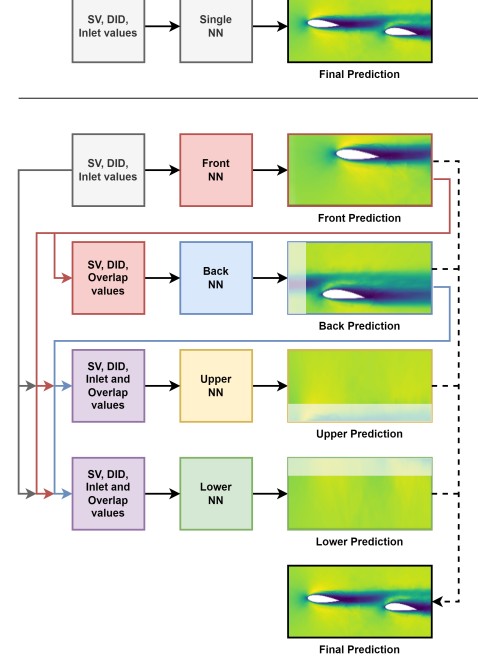

Figure 3: **Above:** Inference using a single NN. **Below:** Inference using a multi-NN

5. All sub-fields are combined to generate the final prediction for the complete flow field, with the latest and most updated prediction used for the overlap regions.

The dependency structure of our multi-NN approach is specifically tied to geometric settings where the division of flow fields aligns with predefined boundaries and inlets. This dependency ensures the sequential propagation of information through sub-domains, maintaining coherence across the overall flow field. For more complex geometries or scenarios with dynamic boundary conditions, this method is adaptable to specific requirements. However, since this technique resonates with with domain decomposition in CFD where a large and complex flow field is broken down into coupled segments (Chan & Mathew, 1994), practitioners in this domain typically can encapsulate the necessary knowledge of inlet positions, overlap regions, and flow directions, thus ensuring that this is rarely a significant limitation in practice.

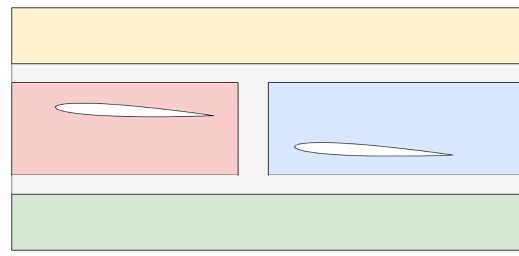

Figure 4: Domain decomposition into front, back, upper and lower fields, with overlapping regions.

Segmentation allows each part to focus on less complex geometries, enhancing prediction accuracy and computational efficiency. Sections like upper and lower fields may be predicted using diverse strategies, such as freestream conditions or interpolation, reducing the total neural networks needed. In some cases, these sections might even be omitted if maintaining the original field size is unnecessary. This adaptability ensures the versatility of the multi-NN inference procedure for diverse CFD applications and specific modelling needs, similar to customisation in domain decomposition based on unique simulation characteristics, such as geometry and boundary conditions (Lim et al., 2023).

## 4 EXPERIMENTS

In this section, we rigorously evaluate the effectiveness of our proposed schemes through three experiments. The primary goal is to assess the performance of the multiple-object DID representation method, evaluate the impact of the pre-training and residual training schemes with an ablation study, determine the effectiveness of multi-NN inference, and finally test the effectiveness of the training schemes in a varying flow condition scenario. We employ four diverse datasets: (i) **Cruise AOA=0°**, (ii) **Cruise AOA=5°**, (iii) **Takeoff AOA=5°**, and (iv) **Cruise Random**. Our method is applied to two state-of-the-art (SOTA) convolutional neural network architectures: MeshGraphNet (MGN) (Pfaff et al., 2021) and invariant edge-GCNN (IVE) (Chen et al., 2021), demonstrating effectiveness across various flow conditions and convolutional types.[3]

### 4.1 DATASETS

Our investigation employs the four-digit series airfoil shapes developed by the National Advisory Committee for Aeronautics (NACA) for aircraft wing applications. The four-digit series NACA airfoils are parameterised using numerical codes ($MPXX$). To comprehensively explore airfoil shapes, parameters $M$ and $P$ are uniformly selected from the range $[0, 6]$, while $XX$ is uniformly chosen from $[5, 25]$ for both single- and tandem-airfoil datasets.

Operating at either a fixed low Reynolds number ($Re = 500$) or a random high Reynolds number, our study considers the interaction between two airfoils interacting, resulting in unsteadiness in the flow fields. For the low Reynolds number condition, two angles of attack (AOA) are considered, specifically $0°$ and $5°$. In contrast, for the random high Reynolds number condition, a range of angles of attack between $-5°$ and $7°$ is considered.

The datasets comprise 1014 single airfoil configurations for Single AOA=0° and Single AOA=5°, and 784 tandem-airfoil configurations for Cruise AOA=0° and Cruise AOA=5°. For 5° AOA, an additional Takeoff AOA=5° dataset with ground effect is utilised. For the Cruise Random dataset,

---

[3]We aim to investigate the effectiveness of our proposed method, which can be used alongside a variety of deep graph architectures architectures. We do not intend to replace any of the original methods.

both $Re$ and AOA are randomly selected within the ranges $\left[10^5, 5\times10^6\right]$ and $[-5°, 7°]$, respectively, totaling 1025 Single- and 900 Cruise-Random cases.

As shown in Fig. 5, the tandem-airfoil datasets are configured using three parameters: gap (G), stagger (S), and height (H) for the Take-off AOA=5 dataset. These parameters are randomly varied within the ranges specified in Tab. 1. For all cases, the training, validation, and test sets were distinct, uniformly sampled sets of 80%, 10% and 10% of the whole.

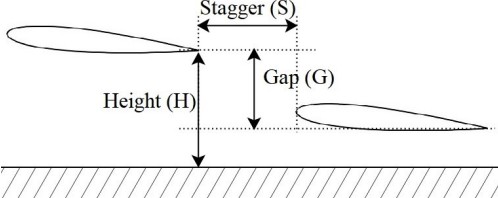

Figure 5: Schematic of tandem-airfoil case.

Table 1: Total number of cases, average grid cells and the range of Reynolds number, $Re$, angle of attack, AOA, stagger, gap and height (see Fig. 5), normalised by the chord length, for the tandem-airfoil datasets

| DATASET | CASES | AVERAGE CELLS | RE | AOA [°] | STAGGER | GAP | HEIGHT |
|---|---|---|---|---|---|---|---|
| CRUISE AOA=0,5 | 784 | $351,315$ | 500 | 0 , 5 | $[0.5, 2]$ | $[-0.4, 0.4]$ | N.A. |
| TAKEOFF AOA=5 | 784 | $271,316$ | 500 | 5 | $[0.5, 2]$ | $[-0.2, 0.6]$ | $[0.4, 1]$ |
| CRUISE RANDOM | 900 | $210,181$ | $\left[10^5, 5 \times 10^6\right]$ | $[-5, 7]$ | $[0.5, 2]$ | $[-0.8, 0.8]$ | N.A. |

### 4.1.1 EXPERIMENT 1: MULTI-OBJECT DID EFFECTIVENESS

In the first experiment, we assess the effectiveness of the multiple-object DID representation by comparing MGN performance with and without DID on the Cruise AOA=0° and Takeoff AOA=5° datasets, where the two airfoils define the boundaries for DID calculation. No additional methods are applied. As shown in Tab. 2, using the DID significantly improves performance for both datasets, with over an 80% reduction in MSE test loss for the Cruise AOA=0° case. These results suggest that the DID representation remains effective even when estimated using the smooth-combining method.

TABLE 2: MSE ($\times 10^{-2}$) PERFORMANCE EVALUATION OF DID

| MODEL / DATASET | CRUISE AOA=0° | TAKEOFF AOA=5° |
|---|---|---|
| MGN + SV | $11.51 \pm 5.48$ | $8.17 \pm 3.37$ |
| MGN + SV + DID | $\mathbf{2.03 \pm 1.15}$ | $\mathbf{4.23 \pm 2.01}$ |

### 4.1.2 EXPERIMENT 2: ABLATION STUDY

The second set of experiments assess the effectiveness of pre-training and residual training on both MGN and IVE, across three datasets. We compare the baseline against the following four combinations of the suggested schemes:

- **PRE**: A single-airfoil model is pre-trained, and its weights are used to initialise the networks of the main tandem-airfoil model before training.

- **PRE-FREE + COMB**: Additionally, the freestream conditions is used as an input feature to the pre-trained single-airfoil model, and the combined flow fields from the single-airfoil network is used as input features to the main tandem-airfoil model.

- **RES-FREE + RES-COMB**: The single-airfoil model is residually trained using the freestream conditions as the estimate fields. Likewise, the main tandem-airfoil model is residually trained using the combined flow fields as estimate fields. However, its weights are not initialised with that of the single-airfoil model.

- **PRE-RES-FREE + RES-COMB**: The pre-trained single-airfoil model is freestream-based residually trained. Its weights are used to initialise the main tandem-airfoil model, which is combined field-based residually trained.

The single-airfoil dataset with the same AOA value (0°, 5° or random) as the tandem-airfoil dataset is used. All models including the baselines incorporate both SV and DID. Likewise, multi-NN inference was used in all models. The outcome of this ablation study is shown in Tab. 3.

The comprehensive method with both pre-training and residual training exhibited superiority for both Cruise AOA=5° and Takeoff AOA=5° datasets, with up to 86% and 75% reduction in loss over the MGN and IVE baseline models respectively. Although it was outperformed by other models

in the Cruise AOA=0 dataset, the difference in performance was an order below the improvements over the first two models. Overall, these results strongly suggest that pre-training and combined-field based residual training both individually enhance the performance of models across various scenarios, but most consistently perform best when used in conjunction with one another.

TABLE 3: MSE ($\times 10^{-2}$) PERFORMANCE EVALUATION OF ABLATION STUDY

| MODEL / DATASET | CRUISE AOA=$0°$ | CRUISE AOA=$5°$ | TAKEOFF AOA=$5°$ |
|---|---|---|---|
| MGN (BASELINE) | $2.03 \pm 1.15$ | $2.08 \pm 9.34$ | $4.23 \pm 2.01$ |
| MGN + PRE | $0.61 \pm 0.40$ | $0.72 \pm 0.41$ | $2.00 \pm 1.02$ |
| MGN + PRE-FREE + COMB | $\mathbf{0.39 \pm 0.33}$ | $0.70 \pm 0.37$ | $0.82 \pm 0.38$ |
| MGN + RES-FREE + RES-COMB | $\mathbf{0.39 \pm 0.28}$ | $0.83 \pm 0.53$ | $1.77 \pm 0.70$ |
| MGN + PRE-RES-FREE + RES-COMB | $0.41 \pm 0.29$ | $\mathbf{0.67 \pm 0.39}$ | $\mathbf{0.59 \pm 0.33}$ |
| IVE (BASELINE) | $0.90 \pm 0.74$ | $1.20 \pm 0.46$ | $2.88 \pm 1.34$ |
| IVE + PRE | $1.12 \pm 0.78$ | $0.99 \pm 0.40$ | $2.45 \pm 1.12$ |
| IVE + PRE-FREE + COMB | $0.86 \pm 0.98$ | $0.88 \pm 0.36$ | $0.88 \pm 0.39$ |
| IVE + RES-FREE + RES-COMB | $\mathbf{0.44 \pm 0.31}$ | $0.66 \pm 0.24$ | $0.73 \pm 0.37$ |
| IVE + PRE-RES-FREE + RES-COMB | $0.47 \pm 0.31$ | $\mathbf{0.60 \pm 0.21}$ | $\mathbf{0.71 \pm 0.32}$ |

### 4.1.3 EXPERIMENT 3: MULTI-NN EFFECTIVENESS

In the third experiment, we assess the effectiveness of the multi-NN inference by comparing MGN performance in a single-NN versus a multi-NN setup, where the field is split into front and back sub-fields. The upper and lower fields were excluded due to memory limitations. The experiment uses the Cruise AOA=$0°$ dataset. As detailed in Tab. 4, multi-NN inference outperforms the single-NN in both models. This suggests that using separate and specialised NN predictions enhances accuracy.

TABLE 4: MSE ($\times 10^{-2}$) PERFORMANCE EVALUATION OF MULTI-NN SCHEME

| MODEL / INFERENCE SCHEME | SINGLE-NN | MULTI-NN |
|---|---|---|
| MGN + RES-FREE + RES-COMB | $1.66 \pm 1.71$ | $\mathbf{1.34 \pm 1.52}$ |
| MGN + PRE-RES-FREE + RES-COMB | $1.51 \pm 1.61$ | $\mathbf{0.60 \pm 0.80}$ |

### 4.1.4 EXPERIMENT 4: EFFECTIVENESS IN VARYING FLOW CONDITIONS

In the final experiment, we assess the method's ability to predict flow fields under varying conditions using the Cruise Random dataset. We test two data sampling styles: Uniform sampling, and Extrapolation. In Extrapolation, the data reflecting the top and bottom 5% of either the AOA or $Re$ value range is used as the test set, while the training and validation sets are uniformly sampled from the middle 90% range. Table 5 shows that the MSE test losses are higher than in other datasets, which is expected due to the varied Reynolds numbers and AOA values without an increase in dataset or model size. Despite this, the proposed method achieves up to 96% reduction in MSE test loss compared to the baseline, demonstrating its effectiveness.

TABLE 5: MSE PERFORMANCE EVALUATION USING CRUISE RANDOM DATASET

| MODEL / DATA SCHEME | UNIFORM | AOA EXTRAP | $Re$ EXTRAP |
|---|---|---|---|
| MGN (BASELINE) | $1.75 \pm 1.35$ | $2.41 \pm 2.26$ | $4.70 \pm 1.60$ |
| MGN + PRE-RES-FREE + RES-COMB | $\mathbf{0.07 \pm 0.07}$ | $\mathbf{0.25 \pm 0.32}$ | $\mathbf{0.26 \pm 0.40}$ |

## 5 CONCLUSION

This paper introduces a novel smooth-combining technique for DID representation in multi-object fields, a pre-training and residual training procedure based on freestream and combined fields, and a multi-NN inference scheme. We also provide four tandem-airfoil datasets for future research on complex geometries. Our methods were rigorously evaluated on two SOTA benchmarks, demonstrating their effectiveness. Overall, our research offers a pioneering methodology that enhances predictive modelling in complex flow fields, providing a more robust and efficient approach with broad implications for CFD and engineering design. This study centres on tandem-airfoil as a complex geometry. Future work should further explore the method's generalisation capabilities in scenarios like multi-airfoil cases in turbomachinery stages, bluff body configurations, or 3D cases.

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

## A  DATABASE GENERATION

**Mesh Generation**  Various meshing methods were employed to cater to different simulation requirements. For simulations involving single airfoil geometries, the *blockMesh* utility tool provided by OpenFOAM-v2112 (Jasak et al., 2007) was utilised to create a C-grid type of hexahedral mesh, encompassing a domain size of 20 chord lengths. In contrast, overset meshing (Benek et al., 1986) was adopted for the simulations involving double airfoil configurations, offering flexibility in manipulating the orientation of the airfoils. The overset mesh consists of a background mesh and one or more component meshes. In our work, the background mesh, which consists of the structure grids, was generated using *blockMesh* with a domain size of $20 \times 24$ chord-length units. On the other hand, the component meshes, which represent the near-field airfoil meshes, were created using Pointwise software (Karman & Wyman, 2019). To ensure accurate resolution of the boundary layer without relying on wall functions, the thickness of the first cell adjacent to the airfoil wall was set to 1 mm, ensuring $y^+ << 1$. Consequently, the single airfoil meshes consist of approximately $1.2 \times 10^5$ cells, while the double airfoil overset meshes comprise a total of approximately $2.7 \times 10^5$ cells, with $2.4 \times 10^5$ cells in the background mesh and roughly $1.5 \times 10^4$ cells in each of the airfoil meshes. These refined meshes enable comprehensive and accurate simulations of the flow behaviour around single and double airfoil configurations, facilitating detailed analysis of their interactions.

**Dataset Generation**  This study considers an extra low Reynolds number ($Re$) scenario, wherein the interaction between two airfoils results in unsteadiness in the flow fields. The ability to predict such unsteadiness in the flow holds significant potential for reducing computational requirements while achieving reliable field initialisation. To this end, two angles of attack (AOA), namely $0°$ and $5°$, are considered. For each AOA, the datasets consist of 1014 single airfoil configurations and corresponding 784 tandem airfoil configurations each with and without ground effect, comprising a diverse combination of different airfoil shapes, ranging from symmetric to asymmetric and thin to thick airfoils. While the inflow conditions remain fixed, variations in geometry shapes are introduced to examine flow interactions between different geometries.

The simulations are conducted at a Reynolds number of $Re = 500$ using steady-state RANS solvers. For single airfoil simulations, the *simpleFoam* solver is employed, while the *overSimpleFoam* solver is utilised for tandem airfoil configurations. Both solvers employ a semi-implicit method for pressure-linked equations (SIMPLE) algorithm (Caretto et al., 1973), coupled with two transport equations from the $k$-$\omega$ SST turbulence model (Menter et al., 2003). Convergence is attained by iteratively solving for $U$, $p$, $k$ and $\omega$ fields until the prescribed convergence criteria are met. The computational resources for this investigation involve the utilisation of 64 CPU cores of the AMD EPYC$^{\text{TM}}$7713 system, accessible through the National Supercomputing Centre, Singapore (`https://www.nscc.sg`).

The dataset generation was inspired by other cases with tandem airfoils from Fanjoy et al. (1997); Yin et al. (2021); Hosseini et al. (2022), and cases with ground effect from Mokhtar (2005); Qu et al. (2015); Grabis & Agarwal (2019). With a total of 4380 cases, including 2352 tandem airfoil cases, this is, to the best knowledge of the authors, the largest public dataset containing the full flow fields around tandem shapes for machine learning purposes. For experiments, the training set, validation set, and test set were distinct, uniformly sampled sets of $80\%$, $10\%$ and $10\%$ of the whole. However, for extrapolation experiments, the data reflecting the top and bottom $5\%$ of either the AOA or $Re$ range was used as the test set. The training and validation sets were uniformly sampled sets of a 8:1 ratio of the remaining middle $90\%$ range of data.

## B  INPUT AND NEURAL NETWORK PARAMETERS

In all experiments, the angle segments $\left(\theta_j, \theta_j'\right) \in \left[\left(-\frac{\pi}{8}, \frac{\pi}{8}\right), \left(\frac{\pi}{8}, \frac{3\pi}{8}\right), \ldots, \left(\frac{13\pi}{8}, \frac{15\pi}{8}\right)\right]$ were used to calculate the DID estimates. These were chosen to give 8 arcs each spanning $\frac{\pi}{4}$ degrees, centred at $\frac{\pi}{4}$ intervals. Also, the maximum distance value used was $d_{max} = 5$.

Table 6 shows the neural network model parameters and training parameters used in the experiments. The input features (and their sizes) to all models consist of the node positions (2), SV (2), DID (8), inlet/overlap values (3) and freestream/combined field (3) when residual training was done, making the input layer size 15 without residual training and 18 with. Likewise, the number of output features

Table 6: Neural network parameters

|  | MGN | IVE |
|---|---|---|
| NUMBER OF HIDDEN LAYERS | 15 | 8 |
| HIDDEN LAYER SIZE (NODE) | $[128, 128, \ldots, 128]$ | $[128, 256, \ldots, 256, 128]$ |
| HIDDEN LAYER SIZE (EDGE) | $[128, 128, \ldots, 128]$ | $[64, 128, 256, \ldots, 256]$ |
| LOSS FUNCTION | MSE | MSE |
| OPTIMIZER | ADAM | ADAM |
| LEARNING RATE SCHEDULER | LAMBDA DECAY | LAMBDA DECAY |
| LAMBDA FUNCTION | $(1 + k \cdot \lambda_0)^{-1}$ | $(1 + k \cdot \lambda_0)^{-1}$ |
| INITIAL LEARNING RATE ($\lambda_0$) | 5.0E-05 | 2.0E-04 |

of all models was 3, for the x-velocity, y-velocity and pressure fields. We use the standard mean squared error (MSE) loss function and Adam optimizer to train the neural networks. A custom decay function is used for the learning rate, as defined in Tab. 6. All models were trained using half-precision and distributed data parallel with the number of GPUs as specified in Tab. 7.

Table 7: Number of GPUs used in parallel

| DATASET \ MODEL | MGN | IVE |
|---|---|---|
| CRUISE AOA=0° | 8 | 4 |
| CRUISE AOA=5° | 8 | 4 |
| TAKEOFF AOA=5° | 7 | 4 |
| CRUISE RANDOM | 4 | - |

## C    SIMULATION AND PREDICTION TIMINGS

This section compares the average wall time required to simulate a flow scenario compared to predicting the flow fields using a neural network. Table 8 shows the average wall time per simulation for each dataset used in training and testing. Note that, each simulation runs in parallel with 64 CPUs. Likewise, Tab. 9 shows the average time per double-airfoil case for each step involved in the neural network prediction using the MGN + pre-free-res + res-combine model.

Table 8: Average simulation timings

| NUMBER OF AIRFOILS | DATASET | SIMULATION TIME (S) |
|---|---|---|
| SINGLE AIRFOIL | CRUISE AOA=0° | $304.42 \pm 32.59$ |
|  | CRUISE AOA=5° | $152.64 \pm 14.99$ |
|  | **AVERAGE** | **228.53** |
| DOUBLE AIRFOIL | CRUISE AOA=0° | $252.02 \pm 56.63$ |
|  | CRUISE AOA=5° | $284.93 \pm 127.49$ |
|  | TAKEOFF AOA=5° | $292.96 \pm 246.70$ |
|  | **AVERAGE** | **276.64** |

Note that the single-airfoil cases are meshed using the standard C-grid mesh, while an overset or "chimera" mesh is used in the double-airfoil cases. The background mesh uses a rectangular mesh and the overset mesh uses a handcrafted mesh. The background mesh cells are then connected to their nearest neighbour cells in the overset mesh to avoid importing two disjointed graphs. Hence, importing a double-airfoil mesh takes longer than a single-airfoil mesh.

From Tabs. 8 and 9, we can see that the average total prediction time comes up to only 65.57 seconds. This is a 76% reduction from the average wall time of 276.64 seconds it takes to simulate a double-airfoil case using OpenFOAM in parallel.

Table 9: Average neural network prediction timings

| Stage | Operation | Simulation Time (s) |
|---|---|---|
| Single Airfoil Predictions | Read CFD file
Calculate Geometric Features ($\times 2$)
Inference ($\times 2$) | $12.00 \pm 0.72$
$3.34 \pm 0.27$
$0.30 \pm 0.00$ |
| Double Airfoil Prediction | Read CFD file + Process Overset Mesh
Calculate Geometric Features
Combine Fields
Inference | $18.97 \pm 0.72$
$17.03 \pm 1.02$
$8.60 \pm 0.68$
$1.10 \pm 0.00$ |
| | Export CFD file
Total | $4.22 \pm 0.39$
$\mathbf{65.57 \pm 3.80}$ |

## D  Training Schemes and Timings

This section presents the average time required by the various NNs in the multi-NN model to train. Note that while the NNs had a maximum epoch of 300, an early-stopping mechanism was utilised, such that training would cease if the validation loss did not improve after 20 epochs, indicating convergence. Additionally, the MGN front models would have a minimum epoch of 150 to ensure sufficient training, due to having more turbulent validation losses.

The training and validation loss curves as shown in Fig. 6 for each sub-domain: (a) front, (b) back, (c) upper, and (d) lower, exhibit consistent convergence. For the front and back sub-domains, the validation losses closely follow training losses across 200 epochs, indicating good generalization to unseen data. Similarly, for the upper and lower sub-domains, both losses converge rapidly and remain stable, suggesting no signs of overfitting. It is worth noting that the size of the computational domain (graph) is large. Prior to domain decomposition, the model may have been underfitting, and this structural decomposition allows the networks to effectively capture sub-domain-specific flow features without increasing the risk of overfitting. These results collectively demonstrate that our approach maintains a balance between model complexity and generalization.

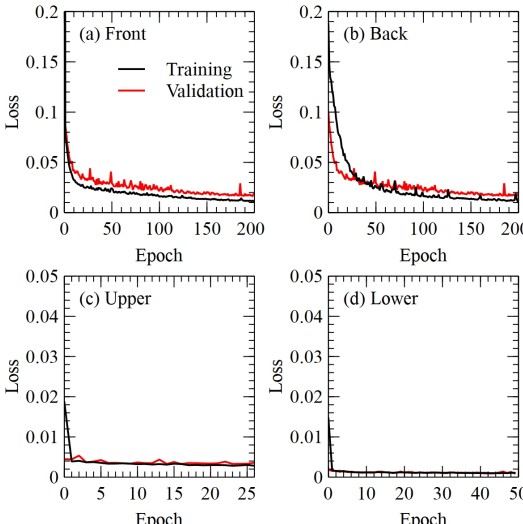

Figure 6: Training and validation loss curves of the various sub-domains.

The average training times of the MGN models using the Takeoff AOA=$5°$ dataset are shown in Tab. 10. The single-airfoil models had larger training set sizes, leading to longer training times. Likewise, due to the simplicity of the upper and lower fields, the upper and lower NNs take the least training time to converge. Hence, these categories are separated.

Table 10: Average neural network training timings

| NN | FEATURES | AVERAGE TRAINING TIME (S) | |
|---|---|---|---|
| SINGLE-AIRFOIL | MGN (BASELINE) | 27,067 | |
| | MGN + FREE | 28,020 | 21,883 |
| | MGN + RES- FREE | 10,562 | |
| FRONT AND BACK | MGN (BASELINE) | 10,452 | |
| | MGN + PRE-FREE | 15,739 | |
| | MGN + PRE-FREE + COMB | 14,056 | 14,517 |
| | MGN + RES-FREE + RES-COMB | 11,878 | |
| | MGN + PRE-RES-FREE + RES-COMB | 20,459 | |
| UPPER AND LOWER | MGN (BASELINE) | 2,853 | |
| | MGN + PRE-FREE | 2,644 | |
| | MGN + PRE-FREE + COMB | 1,905 | 2,191 |
| | MGN + RES-FREE + RES-COMB | 1,770 | |
| | MGN + PRE-RES-FREE + RES-COMB | 1,782 | |

## E  A DISCUSSION ON THE DID

In this section, we will discuss the challenges of calculating the DID as done in the original work and present the justification for using the smooth-combine method to estimate it instead. As mentioned previously, calculating the DID in a multiple-object scenario using Alg. 1 posed certain challenges that were highlighted in red. These challenges are illustrated in Figs. 7 and 8.

The first challenge is in determining whether the point on the object boundary $k$ is obstructed from the point of reference $i$. As shown in Fig. 7(a), in a single object scenario, it suffices to ascertain that either boundary face adjacent to $k$ is on the side of the object that faces $i$. However, as seen in Fig. 7(b), there is the possibility that $k$ is obstructed from $i$ by the boundary faces of another object. Determining obstruction is a process that increases in complexity with the addition of every object.

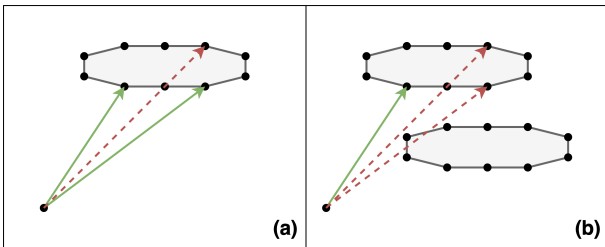

Figure 7: Determining obstruction of a boundary point from the reference point in a (a) single-object case and (b) double-object case. Note how a boundary point that is unobstructed in the first case may be obstructed by another object in the second case.

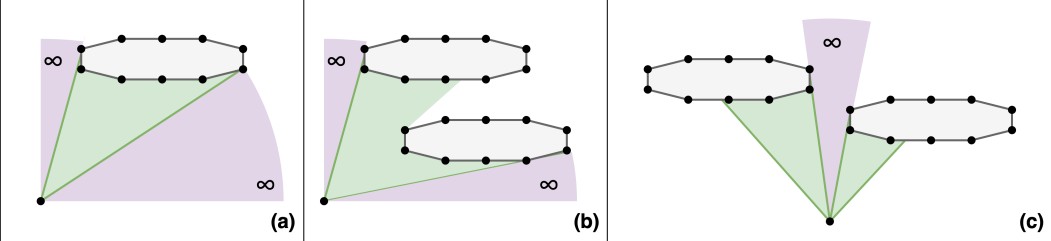

Figure 8: Determining the angular range that faces an object boundary (shown in green) in a (a) single-object case, (b) overlapping double-object case, and (c) non-overlapping double-object case. Note that the angle segment $(\theta_j, \theta_j')$ used was $(0, \pi/2)$ in (a) and (b) and $(\pi/4, 3\pi/4)$ in (c).

Likewise, the second challenge is in determining the proportion of the angular range $(\theta_j, \theta_j')$ where $i$ is obstructed by an object. As shown in Fig. 8(a), in a singular object scenario, this proportion

can be represented as one continuous segment using the minimum and maximum value of $\theta_{i,k}$, the angle at which $k$ is with respect to $i$. However, in a double object scenario, this proportion may be represented as one continuous segment as seen in Fig. 8(b), or two separate segments as seen in Fig. 8(c). Multiple objects involve pair-wise comparisons of each object in determining whether they overlap (as in the former case) or not (as in the latter case), greatly increasing complexity.

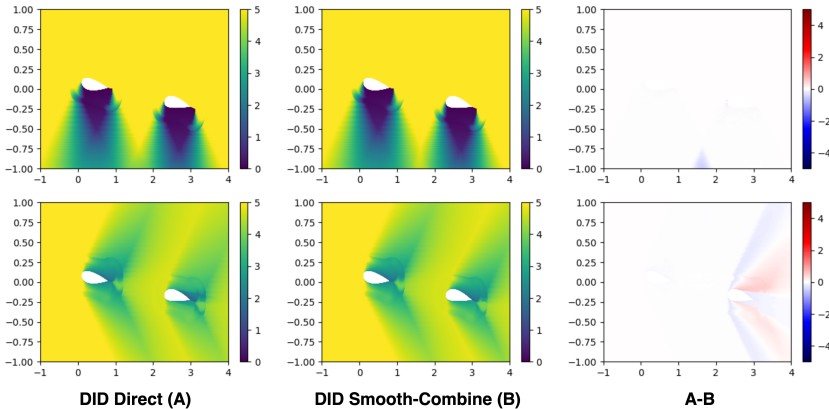

**DID Direct (A)**      **DID Smooth-Combine (B)**      **A-B**

Figure 9: Comparison of the DID field calculated directly with that from smooth-combined estimate. **Above:** Angular range is $[0, \pi]$. **Below:** Angular range is $[\pi/2, 3\pi/2]$.

To circumvent these challenges, a smooth-combining method using the deviation from the maximum value $d_{max}$ was utilised to estimate the DID fields for multiple objects in this paper. While there is a difference between the resulting smooth-combined fields from the direct DID calculation, it is important to highlight that these differences are minimal. To illustrate this, the DID fields for two angle segments from a direct calculation and smooth-combining, as well as their difference, are shown in Fig. 9.

As can be seen from the figures, the most significant difference occurs in points $i$ when both objects are within its angular range. In these areas, the smooth-combination will overestimate the DID when the objects do not overlap (blue regions), and underestimate the DID when the objects overlap (red regions). On the other hand, when only one object is in the angular range of $i$, both the direct numerical calculation and the smooth-combined calculation calculates its average distance to every unobstructed boundary point $j$ on the objects using a relatively uniform weight. Hence, there is little difference between the two. Importantly, the smooth-combined calculation produces a close estimate without harsh lines.

The timings and maximum memory usage of the DID calculation done directly was estimated using a small sample of the datasets. It is compared against the estimation using the smooth-combine scheme in Tab. 11. Note that the direct calculation was not optimised, and some steps that could be done in parallel were instead done in succession. Doing them in parallel would decrease the timing but increase the memory overhead.

Table 11: Comparison of DID calculation times and memory overhead

| CALCULATION TYPE | AVG. TIME PER FIELD (S) | MAX. MEMORY USAGE (GB) |
|---|---|---|
| DIRECT | 5221.641 | 25.5 |
| WITH SMOOTH-COMBINE | 3.492 | 23.3 |

As can be seen, using the smooth-combined estimate saves a significant amount of calculation time for a similar amount of memory required, making it the ideal choice. For a dataset of size $784$, the direct DID calculations would take an estimated $47$ days. The accuracy performance of the direct DID is hence irrelevant when the goal is to produce faster results than numerical simulations.

# F SENSITIVITY STUDIES ON THE DID

To ensure a fair comparison with the baseline methods, hyperparameters such as learning rate, network depth, and layer sizes were kept consistent with the baseline settings. This minimizes variability and ensures that the observed improvements are due to our proposed approach rather than hyperparameter tuning. However, sensitivity to certain domain-specific parameters, such as the maximum DID distance, $d_{max}$ and the number of angle segments used in DID computation, could impact performance. These parameters influence the granularity of the directional distance representation and its ability to capture relevant physical interactions. Sensitivity studies were conducted using the model MGN+PRE-RES-FREE+RES-COMB on the Cruise AOA= $5°$ dataset. The results, as shown in Tab. 12, reveal the impact of varying $d_{max}$ and the number of angle segments on MSE.

TABLE 12: MSE PERFORMANCE EVALUATION FOR SENSITIVITY STUDIES ON DID PARAMETERS.

| $d_{max}$ | NO. OF ANGLE SEGMENTS | MSE($\times 10^{-2}$) |
|---|---|---|
| 5 | 8 | 0.67±0.39 |
| 2.5 | 8 | 0.85±0.38 |
| 5 | 4 | 0.89±0.44 |
| 5 | 16 | 0.51±0.29 |

Reducing $d_{max}$ to 2.5 increases MSE, likely because a smaller $d_{max}$ limits the model's ability to capture longer-range interactions. Similarly, decreasing the number of angle segments to 4 also leads to higher errors, suggesting that fewer angle segments reduce the directional resolution of the DID representation. In contrast, increasing the number of angle segments to 16 improves the performance at the expense of higher computation time for the DID features. Compared with the baseline MGN, whose MSE is $2.08 \times 10^{-2}$ on the same Cruise AOA= $5°$ dataset, the variations of these results are relatively minor, indicating the proposed method's robustness and insensitivity to these parameters.

# G FEASIBILITY OF INDIVIDUAL DID OF EACH OBJECT

Computing a single DID for both objects simultaneously is primarily a practical decision aimed at improving efficiency and scalability. While Alg. 1 can technically compute a single DID for multiple objects simultaneously, its numerical complexity increases significantly with each additional object, resulting in slower training and inference speeds. For instance, Alg. 1 required 5222 seconds to compute the DID for tandem airfoils, whereas Alg. 2, which computes separate DIDs for individual airfoils and then combines them, completed the task in only 3.5 seconds.

Additionally, calculating separate DIDs for each object would increase the input size proportionally to the number of objects. If the dimension of a single object's DID is $N$ and there are $M$ objects, the total input size would scale as $N \times M$, leading to higher memory requirements and computational load on GPUs. By combining the DIDs into a single representation, our approach maintains scalability and significantly reduces computational overhead. Algorithm 2 strikes an effective balance, allowing for efficient handling of multi-object scenarios without sacrificing performance, as discussed in Appendix E.

To further evaluate the feasibility of using individual DIDs for each object, an experiment was conducted to compare the performance and resource usage of individual DIDs versus a single combined DID using the model MGN+PRE-RES-FREE+RES-COMB on the Cruise AOA= $5°$ dataset. The results, as tabulated in Tab. 13, reveal that the single combined DID achieves better computational efficiency and prediction accuracy.

TABLE 13: PERFORMANCE EVALUATION OF EXPERIMENT USING SINGLE COMBINED AND INDIVIDUAL DID ON CRUISE AOA= $5°$ DATASET.

| METHODS | AVERAGE GPU MEMORY USAGE (GB) | MSE ($\times 10^{-2}$) |
|---|---|---|
| SINGLE COMBINED DID | 16.64 | **0.67±0.39** |
| INDIVIDUAL DID | 23.37 | 0.80±0.42 |

## H   SMOOTH-COMBINING METHOD VALIDATION

To validate the effectiveness of the smooth-combining method, we compared its performance against freestream and a simple linear interpolation weighted by the distance to each airfoil as defined,

$$\widetilde{\boldsymbol{U}}(i) = \boldsymbol{\gamma}(i) \cdot \boldsymbol{U}_1(i) + \big(1 - \boldsymbol{\gamma}(i)\big) \cdot \boldsymbol{U}_2(i) \ ,$$

$$\boldsymbol{\gamma}(i) = \frac{d_2(i)}{d_1(i) + d_2(i)} \ , \tag{3}$$

where $d_1$ and $d_2$ are the shortest distances to front (leading) and back (trailing) airfoils, respectively. Figure 10 illustrates the weighting field, $\boldsymbol{\gamma_1}$, generated from distance-based linear interpolation, showing a smooth gradient between the two airfoils. The comparison between (a) freestream, (b) distance-based linear interpolation, and (c) smooth-combining methods is presented in Fig. 11, which shows the absolute error contours of the combined velocity components and pressure fields relative to their corresponding ground truths. The smooth-combining method demonstrates the lowest errors, particularly in the downstream and flow interaction regions, where the freestream and linear interpolation methods show pronounced inaccuracies.

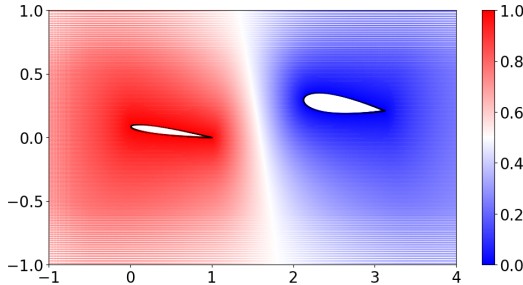

Figure 10: Distance-based linear interpolation weight values for the front airfoil, $\boldsymbol{\gamma_1}$.

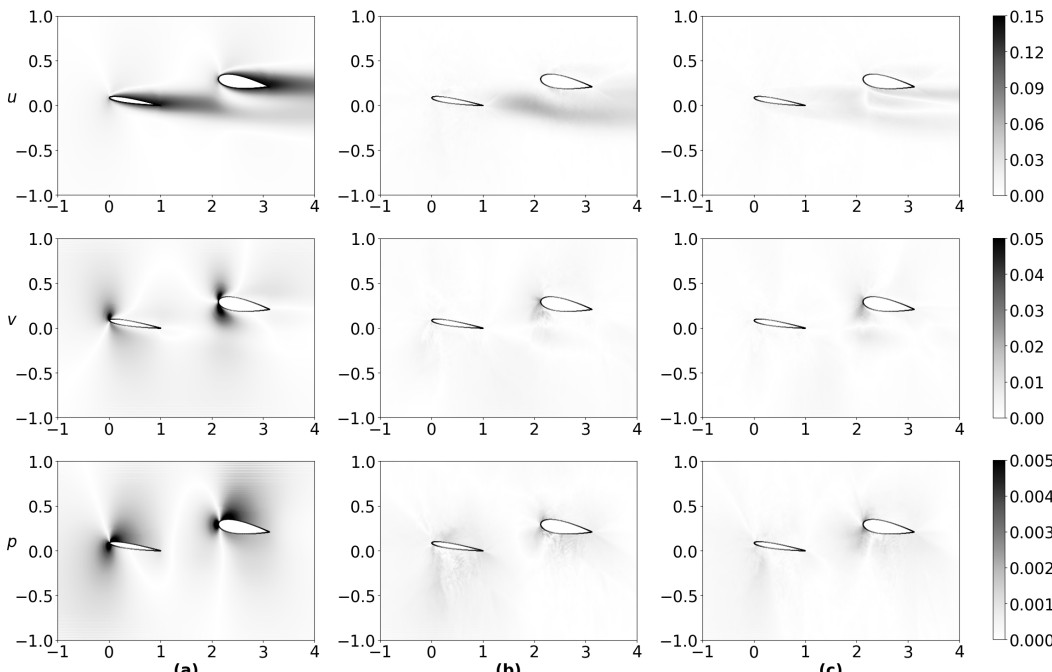

Figure 11: Absolute error contours of combined flow field variables (top row) $u$, (middle row) $v$, and (bottom row) $p$ via (a) freestream, (b) distance-based linear interpolation, and (c) smooth-combining with respect to ground truth $u$, $v$, and $p$.

This qualitative observation is supported by the quantitative results in Tab. 14, where the smooth-combining method achieves the lowest mean absolute error (MAE) with respect to ground truths

across all evaluated metrics, including velocity components and pressure. Specifically, the smooth-combining method outperforms both freestream and linear interpolation with an overall MAE $(\times 10^{-3})$ of $1.46 \pm 0.31$, compared to $1.76 \pm 0.23$ and $6.45 \pm 0.53$ for linear interpolation and freestream, respectively.

TABLE 14: MAE OF COMBINED FLOW FIELDS VIA VARIOUS METHODS AGAINST GROUND TRUTHS.

| METHODS / MAE | $u$ $(\times 10^{-2})$ | $v$ $(\times 10^{-3})$ | $p$ $(\times 10^{-4})$ | OVERALL $(\times 10^{-3})$ |
|---|---|---|---|---|
| FREESTREAM | $1.57 \pm 0.13$ | $3.08 \pm 0.28$ | $5.45 \pm 0.58$ | $6.45 \pm 0.53$ |
| LINEAR INTERPOLATION | $0.39 \pm 0.05$ | $1.14 \pm 0.15$ | $1.95 \pm 0.25$ | $1.76 \pm 0.23$ |
| SMOOTH-COMBINING | $\mathbf{0.31 \pm 0.07}$ | $\mathbf{1.07 \pm 0.16}$ | $\mathbf{1.71 \pm 0.31}$ | $\mathbf{1.46 \pm 0.31}$ |

To further assess the utility of smooth-combining, we conducted an additional experiment using the linear-interpolated flow fields as initial estimators for training the model (MGN + PRE-RES-FREE + RES-COMB) on the Cruise AOA=$5°$ dataset. As shown in Tab. 15, the smooth-combining approach results in significantly lower MSE than linear interpolation. These results confirm that smooth-combining not only provides a more accurate starting point for further training, but also captures complex flow interactions more effectively than alternative approaches. Its superior performance in both initial approximation and subsequent training highlights its importance for this framework.

TABLE 15: MSE $(\times 10^{-2})$ PERFORMANCE EVALUATION OF EXPERIMENT USING LINEAR INTERPOLATED AND SMOOTH-COMBINED FLOW FIELD VARIABLES.

| METHODS / DATASETS | CRUISE AOA=$5°$ |
|---|---|
| LINEAR INTERPOLATION | $1.15 \pm 0.58$ |
| SMOOTH-COMBINING | $\mathbf{0.67 \pm 0.39}$ |

## I ANALYSIS OF PREDICTED FLOW FIELDS

To showcase the effects of varying distance between the two airfoils and increasing AOA on the accuracy of our NN model, we have crafted Figs. 12 and 13. The initial qualitatively compares the prediction of x-velocity, $\hat{u}$, to the ground truth, $u_{\mathrm{GT}}$, for (a) two closely-separated airfoils (S = 0.5, G = -0.05) with strong influence by the front airfoil on the aft airfoil and (b) two distant airfoils (S = 1.8, G = 0.38) that are just mildly interacting with each other. The latter shows contours of x-velocity for approximately (a) positive and (b) negative AOA extremes considered in this work (i.e., $[-5°, 7°]$). All the cases illustrate that, visually, there are little differences between the ground truths and corresponding predictions, thus verifying the robustness of our NN model for a decent range of separation distance between the two airfoils and AOA.

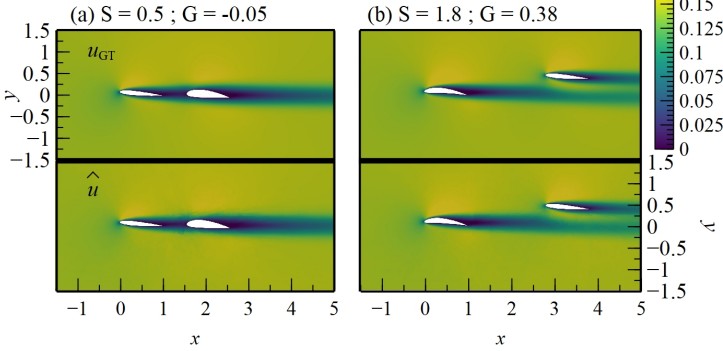

Figure 12: Comparison of ground truth $x-$velocity, $u_{\mathrm{GT}}$ and predicted $\hat{u}$ flow fields by model MGN + pre-res-free + res-comb at (a) closely-separated and (b) distant airfoils for cruise AOA= $5°$ dataset.

To quantify the accuracy of our NN model across different scenarios, we have tabulated the MSE values of our predictions relative to their ground truths under varying Reynolds number, $Re$, AOA, S, and G in Tab. 16. Like the qualitative assessment in Figs. 12 and 13, Tab. 16 confirms once again the consistent robustness of our NN model, with the MSE remaining within a remarkable range of

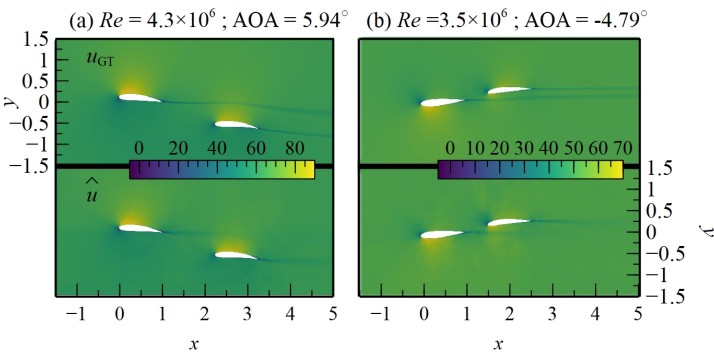

Figure 13: Comparison of ground truth $x-$velocity, $u_{GT}$ and predicted $\hat{u}$ flow fields by model MGN + pre-res-free + res-comb at (a) positive and (b) negative AOA for cruise random datasets.

0.10 (an order smaller than the baseline MSE of 1.75 for the uniform training condition in Tab. 5) regardless of $Re$, AOA, S, and G.

Additionally, we evaluated the normalized residual values of the discrete incompressible Navier–Stokes equations of the SIMPLE algorithm with the predicted x- and y-velocity, $\hat{u}$ and $\hat{v}$, respectively, in Tab. 17. Both variables were predicted with residual that is at least two orders smaller than the maximum value of 1, thus reinforcing the accuracy of our NN as the residual for Navier–Stokes equations is a direct indicator of error relative the exact solution to the simulation (Versteeg & Malalasekera, 2007).

TABLE 16: NORMALIZED MSE OF NN PREDICTIONS UNDER VARYING $Re$, AOA, S, AND G.

| VARIABLE | MSE | | |
|---|---|---|---|
| $Re$ | $< 10^6$ | $10^6 \leq Re < 3 \times 10^6$ | $\geq 3 \times 10^6$ |
| | $0.01 \pm 0.02$ | $0.03 \pm 0.02$ | $0.12 \pm 0.07$ |
| AOA | $< -2°$ | $-2° \leq$ AOA $< 2°$ | $\geq 2°$ |
| | $0.06 \pm 0.07$ | $0.06 \pm 0.06$ | $0.08 \pm 0.07$ |
| S | $< 1.0$ | $1.0 \leq$ S $< 1.5$ | $\geq 1.5$ |
| | $0.07 \pm 0.07$ | $0.06 \pm 0.08$ | $0.06 \pm 0.05$ |
| G | $< -0.4$ | $-0.4 \leq$ S $< 0.4$ | $\geq 0.4$ |
| | $0.06 \pm 0.06$ | $0.06 \pm 0.07$ | $0.07 \pm 0.06$ |

TABLE 17: MEAN AND STANDARD DEVIATION OF NORMALISED RESIDUALS OF DISCRETE INCOMPRESSIBLE NAVIER–STOKES EQUATIONS OF SIMPLE ALGORITHM OVER 10% OF THE RANDOM CRUISE TANDEM AIRFOILS DATASETS.

| VARIABLE | NORMALISED RESIDUAL (MAXIMUM OF 1) |
|---|---|
| $u$ | $0.00203 \pm 0.00017$ |
| $v$ | $0.0168 \pm 0.00074$ |

