# OpenReview forum: "IMPROVING FLOW FIELD PREDICTION OF COMPLEX GEOMETRIES USING SIMPLE GEOMETRIES"
_ICLR.cc/2025/Conference — Submitted to ICLR 2025_

### Official Review · Reviewer_9Pbr · 2024-10-30

**Soundness:** 2
**Presentation:** 2
**Contribution:** 1
**Rating:** 3
**Confidence:** 4

**Summary:**

Using a neural network that has been pre-trained on single-airfoil flow fields, the authors generate flow fields for double-airfoil configurations. The neural network for predicting flow around two airfoils is initialized with weights from the simple-geometry network. The quantity of interest is the velocity magnitude. Visual comparisons and error analysis with reference to the ground truth (i.e., CFD solution) are made.

**Strengths:**

The motivation of this paper is really appreciated and they approached a very important and interesting problem in CFD. They explained everything in detail. The quality of the figures is good. Training time and complexity of the network in terms of a number of trainable parameters are given. The procedure of data generation is explained in detail as well.

**Weaknesses:**

1. This manuscript contains fundamental wrong assumptions about fluid mechanics. I am afraid that the authors are not familiar with the Navier-Stokes equations.

2. The equation 1 does not make sense at all from a physics point of view. How do you justify that from this perspective? Please recall that fluid dynamics is not just a photo that can be easily merged.

3. I disagree with the following statement from the manuscript:
... which suggests that the in fluence of a solid body within a flow field can be conceptualised as deviation from the freestream, Hence, employing weights based on these deviations creates a combined field that effectively preserves the influences of both airfoils, resulting in a close estimate.

4. The five steps described on page 6 do not make sense.

5. This is unsatisfying that the authors first present Alg 1. Later they claim that Alg 1 does not work for some cases, and then the authors proposed Alg 2.

6. The framework predicts only the magnitude of the velocity in 2D, which is not, indeed, a useful quantity in CFD. Instead, we are interested in the velocity component in the x and y directions and the pressure.

**Questions:**

1. The authors claim that the Reynolds number is equal to 500. Hence, the flow is unsteady, in principle. However, in the result, we do not see any time dependency. Did they use RANS or any other averaging method?! This has been totally missed in the manuscript.

2. There is a lot of effort to predict the flow around two airfoils from the solution obtained for one airfoil. The main question is:  Why not simply train the network from scratch for two or three airfoils? Nowadays, neural networks are able to do so and there is no limitation in looking at the literature. Perhaps, the authors wanted to support their claims that their network is able to predict the solution for complex geometries from simple geometries. However, having two airfoils at a relatively large distance is not really a complicated geometric domain.

3. What if we reduce the distance between the two airfoils (referring to Fig. 5 in the manuscript)? Additionally, how would increasing the angle of attack impact the flow prediction? These changes lead to secondary flow, even in the steady-state regime. The question is can the network predict this scenario accurately?

---

> ### Author Response · Authors · 2024-11-22
>
> **Reply to W1**:
> We appreciate the reviewer’s concern regarding the team’s familiarity with the Navier–Stokes equations, but this should not be treated as a weakness of our manuscript because:
> 1. Our core team comprises of an almost equal number of computer scientists and domain experts with doctoral training and years of work experience in aerodynamics, fluid dynamics, and modeling and simulations of (in)compressible/(non-)reacting/turbulent flows.
> 2. Our simulation works have been published in reputable journals for flow physics and numerical methods, while our previous submissions on NN for fluid dynamics have been presented and published in top-tier AI conferences.
>
> Hence, we are hardly strangers to the governing equations and theories of fluid.
> Under this premise, we would appreciate **if the reviewer could specify the “fundamental wrong assumptions about fluid mechanics”** since this is a non-issue internally and, furthermore, our current approach has been reviewed and received support by computational and fluid dynamics specialists of our industry partner, a multinational aerospace engineering company.
>
> **Reply to W2**:
> Thank you for raising this concern. Equation (1) is by design a numerical treatment to provide a close estimate to the ground truth of a tandem airfoil case by reusing its corresponding pair of single airfoil results, which are
> (i) cheaper to generate and
> (ii) more widely available.
> The resultant combined field helps the neural network to focus on key features, making training more efficient. Hence, there is no real value to consider Equation (1) from a physics point of view, though it should be noted that the feasibility of our approach is proven by our better results in terms of a lower MSE than other state-of-the-art methods. So, while it is not meant to be physical technically speaking, Equation (1) does provide an initial condition for learning that bears resemblance to the ground truth.
>
> **Reply to W3**:
> Thank you for the comment. We would appreciate **if the reviewer can elaborate on their disagreement(s) with our statement**, keeping in mind that, as mentioned in our reply to W2, the approach is a numerical treatment to provide a close estimate to the ground truth to ease learning and not meant to be technically physical. That said, from fluid physics perspective, our analogy is not wrong because our configuration of an open (cruise scenario) or semi-open (takeoff scenario) space will generally preserve the uniformity of the freestream if no solid body is embedded in the computational domain. In other words, the resultant flow pattern occurs due to the presence of the solid body in the flow field, thus leading to our statement. Note that we specifically chose the word “deviations” over “perturbations” to account for the possibility that the effects of the solid body may be stronger than small disturbances to the flow, for instance in a fully-developed turbulent wake. Moreover, our approach, which is motivated by our concept, exhibits a lower MSE than other state-of-the-art methods, thus verifying our concept’s usability in practical applications.

---

> ### Author Response · Authors · 2024-11-22
>
> **Reply to W4**:
> Thank you for the feedback. To aid the reviewer’s understanding, we would like to highlight that the five steps described on page 6, which constitute our multi-NN inference procedure, are designed to address the challenges of large-scale complex flow field predictions. Such an endeavor will inherently require domain decomposition to (i) ensure computational tractability and (ii) maximize the benefits of pre-training on single airfoil instances. From a learning point of view, the multi-NN approach is valid, as proven by its better performance than conventional single-NN technique presented in Sec. 4.1.3 of the original (and revised) manuscript. Numerically, we believe that the multi-NN works well because of its alignment with domain decomposition that is common in parallel computational fluid dynamics simulations, especially in the use of overlap regions between adjacent segments of the divided domain. All these points have been covered sufficiently on p. 6, ln. 294 – 298 and p. 7, ln. 324 – 332 of the original manuscript (now on p. 6, ln. 297 – 301 and revised as ln. 333 – 347 on p. 7, respectively, of the revised manuscript).
>
> On a related note, we would like to draw the reviewer’s attention to published NN works like Refs. [1 – 5], which considered only a limited near-field  region of the computational domain that covers the embedded body. Hence, learning from a segment of the domain of a flow field is clearly an accepted practice, and our multi-NN approach should be regarded as an improvement as it enables a scalable handling of a full computational domain that is potentially large to contain multiple geometries. The enhanced capabilities of our multi-NN are due to the directional integrated distance (DID) used, which provides global information to the network to maintain consistency across sub-domain segments.
>
> We hope our reply has made sense of the five steps on p. 6 of the original manuscript for the reviewer. If not, we seek the understanding that our response could only be a speculation on what did not make sense to the reviewer since only a short comment without any justification was given. We would welcome **details on specific aspects of the five steps that the reviewer finds unclear** so as to address the concern(s) more precisely and refine our manuscript if necessary.
>
> References:
> 1. F. Bonnet, J. Mazari, P. Cinnella, P. Gallinari, AirfRANS: High fidelity computational fluid dynamics dataset for approximating Reynolds-averaged Navier–Stokes solutions, Adv. Neural Inf. Process. Syst. 35 (2022) 23463–23478
> 2. O. Obiols-Sales, A. Vishnu, N. Malaya, A. Chandramowliswharan, CFDNet: A deep learning-based accelerator for fluid simulations, in: Proceedings of the 34th ACM International Conference on Supercomputing, ICS ’20, 2020
> 3. D. Kochkov, J. A. Smith, A. Alieva, Q. Wang, M. P. Brenner, S. Hoyer, Machine learning–accelerated computational fluid dynamics, PNAS 118 (2021) e2101784118
> 4. J. Chen, E. Hachem, J. Viquerat, Graph neural networks for laminar flow prediction around random two-dimensional shapes, Phys. Fluids 33 (2021) 123607
> 5. X. Peng, X. Li, X. Chen, X. Chen, W. Yao, A hybrid deep learning framework for unsteady periodic flow field reconstruction based on frequency and residual learning, Aerosp. Sci. Technol. 141 (2023) 108539

---

> ### Author Response · Authors · 2024-11-22
>
> **Reply to W5**:
> Thank you for the comment but, in our opinion, the flow of presentation of Alg. 1 and Alg. 2 is necessary to guide the general readers who may not be familiar with the concept of directional integrated distance (DID) and/or its implementation. In essence, in the original manuscript, we presented the DID concept on p. 2, ln. 74 – 78, followed by how it can be simply but ineffectively implemented as Alg. 1 on p. 2, ln. 83 – 96. Note that the limitation in this direct implementation has been clearly specified on p. 2, ln. 79 – 81, and we have carefully positioned the above all under a “Preliminaries” section (Sec. 2.1). Then, in Sec. 3.2, p. 5, ln. 237 – 240 of the original manuscript, we reiterated the problem of Alg. 1 before proposing on p. 5, ln. 249 – 256 our Alg. 2 to address the said problem, which is one of our novel contributions in this work. Further discussion on the superiority of Alg. 2 over Alg. 1 was given in Appendix E, p. 15 – 16 of the original manuscript. Respectively, the stated excerpts are now found without major modifications on p. 2, ln. 78 – 82; Alg. 1, p. 2, ln. 87 – 100; p. 2, ln. 83 – 85; Sec. 2.1 "Preliminaries"; Sec. 3.2 "DID Calculation for Multiple Objects", p. 5, ln. 241 – 243; Alg. 2, p. 5, ln. 251 – 258; and Appendix E, p. 16 – 17; of the revised manuscript.
>
> We would like to highlight that our presentation style is common in AI papers and logical to us, though we humbly remain open to the reviewer’s **advice on how else we can present our train of thought satisfyingly**.
>
> **Reply to W6**:
> We thank the reviewer for validating the interests of the CFD community in our NN framework, which in fact, contrary to the reviewer's claim, **predicts all primitive variables (i.e., velocity component in the x and y directions and pressure) of the flow field**. In the original manuscript, this predictive capability has been (i) alluded to on p. 4, ln. 209 – 210 with “... such as the x-velocity fields”; (ii) implied throughout Sec. 3.1 with boldface letters like $\mathbf{U}$ and $\mathbf{y}$, which are commonly used in fluid mechanics community to indicate a solution vector of primitive variables (see Ref. [1] for instance); and (iii) stated explicitly in Appendix B, p. 13 – 14, ln. 701 and 715 by “... number of output features ... was 3, for the x-velocity, y-velocity and pressure fields”. Respectively, the stated excerpts are now found without major modifications on p. 5, ln. 218 – 219; Sec. 3.1, "Smooth-Combining"; and Appendix B, p. 13 – 14, ln. 701 and 715; of the revised manuscript.
>
> For these reasons, we are quite confused by the reviewer’s comment that “The framework predicts only the magnitude of the velocity in 2D”, not to mention that **the words “velocity magnitude” have never been used in our original (and revised) manuscript**.
>
> Reference:
> 1. H. Oertel and L. Prandtl, Navier–Stokes Equations. In Prandtl–Essentials of Fluid Mechanics, S. Antman, J. Marsden, and L. Sirovich (Eds.). Springer Science+Business Media, LLC, New York, NY 10013, USA, (2010) 287 – 291

---

> ### Author Response · Authors · 2024-11-22
>
> **Reply to Q1**:
> Thank you for the question. We agree that, at Reynolds number of 500, the airfoil configuration is most likely in a largely laminar but unsteady regime. However, our datasets were generated with the Reynolds-averaged Navier–Stokes (RANS) methodology, so there will not be any time dependency in both the ground truths and predictions. Essentially, our framework is currently built to predict the steady mean flow variables. The reviewer’s comment, “Did they use RANS or any other averaging method?! This has been totally missed in the manuscript.”, is in fact inaccurate because we have clearly written in Appendix A, p. 13, ln. 675 – 676 of the original (and revised) manuscript that “The simulations are conducted at a Reynolds number of Re = 500 using steady-state RANS solvers.”
>
> We also would like to stress that the absence of a description of our numerical setup in the main text is a conscious decision by us to meet the page count constraint but without loss of generality, since:
> 1. The overarching objective of our method is to predict the simulation results as closely as possible, so information like the use of a RANS solver is irrelevant, especially to readers from the AI community.
> 2. We are confident of the physical validity of our benchmark datasets, having accounted for the necessary grid and residual convergence, as mentioned in Appendix A, p. 13, ln. 679 – 681 of both original and revised manuscript, and verified the simulation results with that of existing literature. Therefore, attaining the previous point also means that our approach does follow flow physics, thus fulfilling the physics-informed aspect of the NN developed in this work.
>
> **Reply to Q2**:
> We thank the reviewer for the opportunity to reemphasize our work’s objective by this question. As mentioned in various parts of the original (and revised) manuscript, e.g., in the Abstract, p. 1, ln. 14 – 17 and Introduction, p. 1, ln. 40 – 43, our intention is to prove that learning of multi-body scenarios, which can be convoluted with interactions between the bodies, can benefit from reusing simpler single-body results, which are typically in abundance.  In contrast, complex multi-body datasets are much less available, true even for the arguably canonical case of tandem-airfoil, which we have chosen to study in this work for its widespread application in various engineering domains (cf. p. 1, ln. 43 – 47 of the revised manuscript, or p. 1, ln. 43 – 48 of the original manuscript, for details). To the best of our knowledge, we would be the first group to have generated four tandem-airfoil datasets with over 2000 CFD simulations, which we are keen to share for future exploration once this work is published, as stated on p. 2, ln. 54 – 56 and ln. 67 – 68 of the revised manuscript and p. 2, ln. 54 – 55 and ln. 63 – 64 of the original manuscript.
>
> Therefore, the reason to why we put in “... a lot of effort to predict the flow around two airfoils from the solution obtained for one airfoil” is for the fact that there is generally a scarcity of data for training of NN in multi-body scenarios, for which we have demonstrated a possibility to address by the utilization of single-body solutions. Such a concern on data scarcity is common in the AI community, as can be seen from previous works like Refs. [1, 2] that looked specifically into the so-called scarce data regime. Besides, our results presented in Tabs. 3 and 5 of the original (and revised) manuscript indicate that, in terms of the tandem-airfoil configuration, our approach (i.e., those labelled with “COMB”) performs much better than direct learning from multiple bodies (i.e., the “BASELINE” cases).
>
> Regarding the last comment, “... having two airfoils at a relatively large distance is not really a complicated geometric domain”, we would gently remind the reviewer that, as stated explicitly on p. 8, ln. 380 – 387 and Tab. 1 of the revised manuscript or p. 8, ln. 380 – 383 and Tab. 1 of the original manuscript, **a wide range of arrangement of the two airfoils from as close as less than half chord-length (which is not practical for the modeling of tandem-wing aircrafts) to as far as two chord-lengths has been considered in this work, along with variations in Reynolds number, angle-of-attack, and magnitude of ground effect**. Some visual aids have been provided for the reviewer in our reply to Q3, which poses a related query on the reduction of distance between the two airfoils.
>
> References:
> 1. F. Bonnet, J. Mazari, P. Cinnella, P. Gallinari, AirfRANS: High fidelity computational fluid dynamics dataset for approximating Reynolds-averaged Navier–Stokes solutions, Adv. Neural Inf. Process. Syst. 35 (2022) 23463–23478
> 2. A. Bansal, R. Sharma, and M. Kathuria, A Systematic Review on Data Scarcity Problem in Deep Learning: Solution and Applications, ACM Comput. Surv. 54, 10s, (2022) 208

---

> ### Author Response · Authors · 2024-11-22
>
> **Reply to Q3**:
> Thank you for the question as it gives us the opportunity to showcase more results of our NN framework that are not critical for the manuscript, as shown in Figs. 12 and 13 and Tabs. 16 and 17 of the revised manuscript. Before discussing these additional results, we would like to specify that Fig. 5 of the original (and revised) manuscript is supposed to be a schematic that explains the notations S, G, and H for stagger, gap, and height in the tandem-airfoil configuration, respectively. We have simulated a wide range of S, G, and H, as stated explicitly in Tab. 1 of the original (and revised) manuscript, which is in close proximity to Fig. 5 and even makes reference to it in the caption. In other words, Fig. 5 of the original (and revised) manuscript should not be treated as a setup fixed in our work.
>
> Regarding the reviewer’s queries on the effects of varying distance between the two airfoils and increasing angle-of-attack (AOA) on the accuracy of our NN, we have crafted Figs. 12 and 13. The initial qualitatively compares the prediction of x-velocity, $\hat{u}$, to the ground truth, $u_\text{GT}$, for (a) two closely-separated airfoils (S = 0.5, G = -0.05) with strong influence by the front airfoil on the aft airfoil and (b) two distant airfoils (S = 1.8, G = 0.38) that are just mildly interacting with each other. The latter shows contours of x-velocity for approximately the (a) positive and (b) negative AOA extremes considered in our work (i.e., [-5° , 7° ]). All the cases illustrate that, visually, there are little differences between the ground truths and corresponding predictions, thus verifying the robustness of our NN for a decent range of separation distance between the two airfoils and AOA.
>
> To quantify the accuracy of our NN across different scenarios, we have tabulated the MSE values of our predictions relative to their ground truths under varying Reynolds number, $Re$, AOA, S, and G in Tab. 16 of the revised manuscript. Like the qualitative assessment in Figs. 12 and 13 of the revised manuscript, Tab. 16 confirms once again the consistent robustness of our NN, with the MSE remaining within a remarkable range of ~0.10 (an order smaller than the baseline MSE of 1.75 for the uniform training condition in Tab. 5 of the revised manuscript) regardless of $Re$, AOA, S, and G.
>
> Table 16:
>  | Variable | MSE           |                                   |                   |
> |:--------:|:-------------:|:---------------------------------:|:-----------------:|
> | $Re$     | $<10^6$       | $10^6 \leq Re< 3\times10^6$       | $\geq3\times10^6$ |
> |          | $0.01\pm0.02$ | $0.03\pm0.02$                     | $0.12\pm0.07$     |
> | AOA      | $<-2^\circ$   | $-2^{\circ} \leq $ AOA $<2^\circ$ | $\geq 2^\circ$    |
> |          | $0.06\pm0.07$ | $0.06\pm0.06$                     | $0.08\pm0.07$     |
> | S        | $<1.0$        | $1.0\leq$ S $< 1.5$               | $\geq 1.5$        |
> |          | $0.07\pm0.07$ | $0.06\pm0.08$                     | $0.06\pm0.05$     |
> | G        | $<-0.4$       | $-0.4\leq$ S $< 0.4$              | $\geq 0.4$        |
> |          | $0.06\pm0.06$ | $0.06\pm0.07$                     | $0.07\pm0.06$     |
>
> Additionally, we evaluated the normalized residual values of the discrete incompressible Navier–Stokes equations of the SIMPLE algorithm with the predicted x- and y-velocity, $\hat{u}$ and $\hat{v}$, respectively, in Tab. 17 of the revised manuscript. Both variables were predicted with residual that is at least two orders smaller than the maximum value of 1, thus reinforcing the accuracy of our NN as the residual for Navier–Stokes equations is a direct indicator of error relative the exact solution to the simulation [1].
>
> Table 17:
> | Variable | Normalised Residual (Maximum of 1) |
> |:--------:|:----------------------------------:|
> | $u$      | $0.00203\pm0.00017$                |
> | $v$      | $0.0168\pm0.00074$                 |
>
> The above discussion has been included as Appendix I in the revised manuscript to provide further credits to our proposed approach.
>
> Reference:
> 1. H. Versteeg and W. Malalasekera, Multigrid Techniques. In An Introduction to Computational Fluid Dynamics: The Finite Volume Method. Pearson Education Limited, Harlow, Essex, CM20 2JE, UK, (2007) 230 – 231

---

> ### Author Response · Authors · 2024-11-27
> **Gentle Reminder**
>
> Dear reviewer 9Pbr,
>
> We sincerely appreciate the time and effort you have dedicated to reviewing our paper.
>
> As the discussion period approaches its end, we wanted to gently remind you that there are seven days remaining to share any additional comments or questions you might have. Your insights are invaluable, and we would be grateful for the opportunity to address any further concerns or feedback you may have before the discussion phase concludes.
>
> Thank you once again for your valuable contribution to this process.
>
> Best regards,
> Authors

---

> > ### Comment · Reviewer_9Pbr · 2024-11-28
> >
> > Thank you for your response and the effort you’ve put into addressing my concerns.
> >
> > My evaluation is based only on the submitted manuscript, which is reviewed anonymously, as required by ICLR. This ensures fairness and objectivity. You mentioned:
> >
> > *Our simulation works have been published in reputable journals for flow physics and numerical methods, while our previous submissions on NN for fluid dynamics have been presented and published in top-tier AI conferences.*
> >
> > While it’s great that your group has published in top AI conferences, each submission to ICLR is judged on its own merits.
> >
> > Concerning my question 3, my concern has not been resolved. In the added Fig. 12 and Fig. 3, you have chosen two thin airfoils with the angle of a very small angle of attack. Of course, because the first airfoil does not make a strong wake region, then its influence on the second airfoil is minimal. You have chosen these setups smartly, but this is misleading. I encourage you to look at publications in the fluid mechanics community such as those published in Journal of Computational Physics (if you come from the computer science community).
> >
> >  For instance, please take a look at Figure 16 of the following paper:
> >
> > https://doi.org/10.1063/1.5141571
> >
> > Or please take a look at Fig. 12 of the following paper:
> >
> > https://doi.org/10.1115/1.4066641
> >
> > And there are so many others …
> >
> > These are examples, that the second body indeed affects the flow between two bodies. Can your proposed network predict such scenarios?!

---

> ### Author Response · Authors · 2024-11-30
>
> Thank you very much for the response. Following are our replies to the various points that the reviewer has brought up.
>
> **Judging of Submission** —
> We appreciate the reviewer’s emphasis on judging our manuscript solely *on its own merits* and with *fairness and objectivity*, as required by ICLR. However, we would like to highlight that the mentioning of our previous works having been published in both top AI conferences and journals of computational and fluid communities is in response to the reviewer's earlier concern in W1 regarding our credibility:
>
> *I am afraid that the authors are not familiar with the Navier-Stokes equations.*
>
> Besides, we also clarified first that our group has an equal number of fluid domain experts and AI experts, so our intention was not to affect the evaluation of our submission with previous achievements, but to respond transparently to an inaccurate point made about our background.
>
> **Concerns in Q3** —
> For clarity, we copy Q3 here without any modifications:
>
> *What if we reduce the distance between the two airfoils (referring to Fig. 5 in the manuscript)? Additionally, how would increasing the angle of attack impact the flow prediction? These changes lead to secondary flow, even in the steady-state regime. The question is can the network predict this scenario accurately?*
>
> As shown, the question is in regard to *reduce the distance between the two airfoils* and *increasing the angle of attack*, to which we responded with the added Figs. 12 and 13 (Note: We are guessing that the reviewer meant Fig. 13 instead of *Fig. 3*.), respectively. So, thickness of the airfoils and magnitude of the angle of attack in the current reviewer response are really new concerns, not that the concerns in Q3 have not been replied to (see **Reply to Q3** for details). Nevertheless, we shall address these new concerns in the following.
>
> On the remark of *two thin airfoils*, it is clearly untrue because Fig. 12(a) shows a back airfoil with a thickness-to-chord ratio ($t/c$) of 18%, which is not considered thin generally. In fact, even the front airfoil in Fig. 12(a) has a $t/c$ of 10%, which, for reference, is already within the range of thickness of conventional aircraft wings [1] and tandem-airfoil configurations like in Ref. [2] that inspired our dataset generation, as well as typical tube launched UAVs like Switchblade [3]. Interestingly, Ref. [4] that was suggested by the reviewer also considered a NACA 4412 airfoil that has a close $t/c$ of 12%. Frankly, there is no definitive threshold to thin airfoil theory, but Torenbeek [5] did provide a classification based on stalling properties of airfoil that *thin airfoil stall* happens on sections of $t/c\lesssim$ 6%.
>
> Besides, sufficient camber, like the airfoils in Figs. 12(b) and 13, can too affect the wake regardless of airfoil thickness, so it is, in our opinion, shallow to isolate thickness as the only important factor. But we have done so anyway since this is the reviewer's latest concern, with additional context given in the table below to emphasize the performance of our NN in scenarios where the front airfoil has (i) a minimum $t/c$ of 12% and is (ii) thicker than and (iii) vertically close to the back airfoil. Hence, these specific cases are similar to the arrangement in Fig. 12(a) but with equally thick/thicker front airfoil so that, like seen in Fig. 12(a), the back airfoil will reside fully in the wake of the front airfoil where the mean x-velocity is less than 10% of the freestream velocity magnitude. Note that Yin et al. [2] has previously demonstrated that such a tandem-airfoil arrangement will result in noticeably different aerodynamic characteristics of both front and back airfoils than their corresponding single airfoil cases, so the reviewer may have to rethink their point that:
>
> *Of course, because the first airfoil does not make a strong wake region, then its influence on the second airfoil is minimal.*
>
> | Gap (G)    | Cruise AOA=$5^\circ$ || Random ||
> |:-----------:|:-----------------:|:-----------------:|:-----:|:------------:|
> |                 | MSE                  | % of Dataset                | MSE                 | % of Dataset |
> | [-0.2, 0.2] | $0.62\pm0.44$ | 19%                             | $0.05\pm0.08$ | 13%              |
> | [-0.3, 0.3] | $0.50\pm0.36$ | 36%                             | $0.05\pm0.07$ | 20%              |
>
> In the table, we can see that the MSE of our NN remains low and comparable to the overall MSE presented in Tabs. 3 and 5 of the revised (and original) manuscript. Additionally, these statistics are based on a substantial portion of our datasets, thus confirming that our NN consistently performs well even when significant interactions between the two airfoils are present, with Fig. 12(a) being a representative case rather than a cherry-picked example.

---

> ### Author Response · Authors · 2024-11-30
>
> **Concerns in Q3 (Cont'd)** —
> Which brings us to the reviewer's insinuating remark:
>
> *You have chosen these setups smartly, but this is misleading.*
>
> We would like to remind the reviewer that, as stated p. 7, ln. 365 – 367 of the revised manuscript (or p. 7, ln. 360 – 362 of the original manuscript), airfoil $t/c$ was uniformly sampled from 5% to 25%, thus ensuring that both thin and thick airfoils were adequately represented in our datasets. Hence, like aforementioned and supported by the results in the above table, Figs. 12 and 13 are representations of many similar cases in our datasets that were used to address the original Q3, which does not pertain to airfoil thickness. Therefore, there is no effort made to mislead the readers with our choices. We would have generated more images to support this claim, but cannot do so regretfully because the reviewer response came in with only 6 hours left for revisions in the manuscript to be eligibly made, which is too little turnover time, and OpenReview does not allow for media attachment. Still, our integrity can be proven by the fact that our parameters and methods are all transparently presented in the original and revised manuscripts. Moreover, since we intend to share our datasets with the community once our work is published, the influence from the front airfoil on the back airfoil can be checked by other researchers.
>
> With due respect, what is really *misleading* is the reviewer's constant false claims of our work, for example in W6, that
>
> *The framework predicts only the magnitude of the velocity in 2D*,
>
> or in Q1, that
>
> *Did they use RANS or any other averaging method?! This has been totally missed in the manuscript.*,
>
> or here, that
>
> *you have chosen two thin airfoils with the angle of a very small angle of attack*.
>
> The first two examples have been refuted in our **Reply to W6** and **Reply to Q1**, respectively, for which the reviewer has only acknowledged with a simple one liner in the current response. For the last example, other than the inaccuracy of *thin airfoils* that has been addressed earlier, the use of *very small angle of attack* is not correct either. As we have pointed out in our **Reply to Q3**, we have considered an angle-of-attack (AOA) range of $[-5^\circ, 7^\circ]$, which is not *very small* by conventional standard. Granted, the range does not cover the stall regime, but it aligns with existing works like Ref. [2], which we relied on for validation of our datasets. Not to forget that Yin et al. [2] has confirmed that $5^\circ$ for both front and back airfoils is sufficient to bring about different aerodynamic characteristics in tandem-airfoil configurations relative to their single airfoil counterparts.
>
> References:
> 1. M. H. Sadraey, 3.4.2 Wing, Horizontal Tail, and Vertical Tail. In Aircraft Performance: An Engineering Approach. CRC Press, Boca Raton, FL 33487-2742, USA, (2017) 74
> 2. B. Yin, Y. Guan, A. Wen, N. Karimi, and M. H. Doranehgard, Numerical simulations of ultra-low-Re flow around two tandem airfoils in ground effect: Isothermal and heated conditions, J. Therm. Analy. Calorim. 145 (2021) 2063–2079
> 3. I. S. Kryvokhatko and O. M. Masko, Aerodynamic Characteristics and Longitudinal Stability of Tube Launched Tandem-Scheme UAV, K. Volkov (Ed.).In Flight Physics: Models, Techniques and Technologies. InTech, London, W1W 5PF, UK, (2018) 76
> 4. S. Gupta, R. Naveen, B. V. Prathik, and S. Spurthy, CFD Simulation of turbulent flow around multi-element airfoil, AIP Conf. Proc. 2204 (2020) 03008
> 5. E. Torenbeek, 7.3.3. Stalling Properties of Airfoil Sections. In Synthesis of Subsonic Airplane Design. Delft University Press, RT Delft, Holland and Martinus Nijhoff Publishers, The Hague, Holland, (1982) 229 – 231

---

> ### Author Response · Authors · 2024-11-30
>
> **Other Scenarios** —
> We acknowledge the importance of other examples that the reviewer has raised concerns for, specifically Figs. 16 (high-lift devices) and 12 (wall-mounted trapezoidal bluff bodies) of Refs. [1] and [2], respectively. In theory, our framework is applicable to these scenarios since they all can be decomposed into simpler single units, a point that should be intuitive once our method is accepted. (Note: We assume this is indeed the case for the reviewer since there is no further issue posed for W1 – W6 and Q1 – Q2.) However, one should recognize that these scenarios are significantly different from the tandem-airfoil configuration, which has been explicitly stated as our scope of study in the Abstract (p. 1, ln. 17) and Introduction (p. 1, ln. 43 – 47).
>
> Seeing their familiarity with ICLR regulations, the reviewer would know that additional experiments *should not significantly change the content of the submission. Rather, they should be limited in scope and serve to more thoroughly validate existing results from the submission.* So to ask for new experiments based on other scenarios like that of Refs. [1, 2] is not fair, not to mention an intentional omission of our key objective to prove the feasibility to train a NN to provide a close estimate to the ground truth of a tandem airfoil case by reusing its corresponding pair of single airfoil results, which are (i) cheaper to generate and (ii) more widely available, as we have highlighted in our **Reply to W2**.
>
> The reviewer should realize that, to substantiate our approach in this submission, we had generated four tandem-airfoil datasets that comprise of over 2000 simulations under varying parameters like Reynolds number and angles-of-attack. These datasets have to be designed specifically to challenge the robustness and generalizability of our NN framework and are, to the best of our knowledge, the first of their kind tailored for AI-CFD training in terms of tandem-airfoil configuration, as we have emphasized in our **Reply to Q2**. In fact, both Reviewers 1Tte and kz1T have endorsed the importance of our datasets, stating, respectively:
>
> *I think the simulated datasets are also very valuable for the community.*
>
> and
>
> *The paper also introduces a new set of datasets, which would represent a valuable contribution to the community if made available as open-source.*
>
> In other words, the applicability of our NN is contingent on having appropriate training datasets. So we will not be able to quantitatively address the reviewer's concern because datasets for the suggested scenarios are not publicly available nor adequately curated for training purposes. However, we are always happy to expand our work to cover a broader range of cases of interest to the fluid mechanics community, provided that the reviewer can provide the related and appropriate training datasets.
>
> References:
> 1. S. Gupta, R. Naveen, B. V. Prathik, and S. Spurthy, CFD Simulation of turbulent flow around multi-element airfoil, AIP Conf. Proc. 2204 (2020) 03008
> 2. S. Sarkar, N. Gupta, K. Debnath, P. R. L. Raj, Computational analysis of flow around two wall-mounted trapezoidal bluff bodies arranged in tandem position, J. Fluids Eng. 147 (2025) 021301

---

> > ### Author Response · Authors · 2024-11-30
> >
> > In closing, we seek the reviewer's understanding that our submission is meant to be a proof-of-concept for our novel method rather than its implementation over many different configurations. Under this premise and precisely because there are so many scenarios available, there is a need to focus our work. So we chose the tandem-airfoil configuration as it has the appropriate amount of challenges for our endeavor despite being a canonical case. Similar approach has been applied in published works of both computational/fluid journals and AI conferences [1 – 4], thus indicating its acceptance by both fluid and AI communities.  (Note: Reference [1] from the Journal of Computational Physics that the reviewer has specially mentioned looked only at angles-of-attack between $\pm5^\circ$ and airfoils with thickness-to-chord ratio between ~9% to ~12%, as well as a moderate Mach number range of [0.2, 0.6].)
> >
> > Finally, we urge the reviewer to read through our submission and replies more carefully as the concerns and questions so far have mostly been addressed by points that we have already made in the manuscript. Also, we would appreciate if the reviewer can avoid making more unsubstantiated accusations, such as, in W1:
> >
> > *This manuscript contains fundamental wrong assumptions about fluid mechanics.*
> >
> > While we truly appreciate the time that the reviewer has spent, the above behaviors have only prompted us to mundanely reiterate excerpts from our manuscript and precluded any constructive improvements to our submission.
> >
> > References:
> > 1. W. Cao, J. Song, and W. Zhang, Solving high-dimensional parametric engineering problems for inviscid flow around airfoils based on physics-informed neural networks, J. Comput. Phys. 516 (2024) 113285
> > 2. X. Ren, P. Hu, H. Su, F. Zhang, and H. Yu, Physics-informed neural networks for transonic flow around a cylinder with high Reynolds number, Phys. Fluids 36 (2024) 036129
> > 3. F. Mazé and F. Ahmed, Diffusion models beat GANs on topology optimization, in: Proceedings of the 37th AAAI Conference on Artificial Intelligence, 2023
> > 4. F. Bonnet, J. Mazari, P. Cinnella, P. Gallinari, AirfRANS: High fidelity computational fluid dynamics dataset for approximating Reynolds-averaged Navier–Stokes solutions, Adv. Neural Inf. Process. Syst. 35 (2022) 23463–23478

---

> > > ### Comment · Reviewer_9Pbr · 2024-12-02
> > >
> > > Thanks for your comments.
> > >
> > > Given your response and explanation, I can now strongly conclude that the framework you provided has no contributions. This is 2D, predicts flow around objects that have a very limited effect on each other. Given the current computational resources, no one needs such a framework. The easiest way is to simply train a network with multiple bodies and then ask it to predict new scenarios.
> > >
> > > Instead of taking my questions seriously at the beginning of the discussion period, the authors wrote irrelevant, non-scientific comments, such as how much they are experts in AI and how many journal/conference papers they have published in top journals/conferences. They now also argue that they did not conduct the experiments because it would change their submission a lot, which is not true.
> > >
> > > All in all, I keep my rating. The final decision will be made by the chair.

---

> > > > ### Author Response · Authors · 2024-12-02
> > > >
> > > > On the comment:
> > > >
> > > > *authors wrote irrelevant, non-scientific comments, such as how much they are experts in AI and how many journal/conference papers they have published in top journals/conferences*,
> > > >
> > > > as we have mentioned in **Judging of Submission**, our mentioning of our background and past accomplishments were brought up because of the reviewer’s concern in W1, which wrongly discredits our work with:
> > > >
> > > > *This manuscript contains fundamental wrong assumptions about fluid mechanics. I am afraid that the authors are not familiar with the Navier-Stokes equations.*
> > > >
> > > > Note that we have asked the reviewer to specify what are the *fundamental wrong assumptions* in our **Reply to W1**, but the request was ignored. Also, the reviewer seems to have an issue acknowledging the fact that our team is interdisciplinary with a good mix of both AI **and** fluid dynamics experts.
> > > >
> > > > To be clear, there are only two sets of *non-scientific* comments, namely **Reply to W1** and **Judging of Submission**, but they are not *irrelevant* as we were addressing the reviewer’s question over our credibility. All other responses are technical and relevant and, to support this claim, we provide an outline in the following:
> > > > - **Reply to W2** — To explain that Equation (1) is by design a numerical treatment, which was commented by the reviewer that it *does not make sense at all from a physics point of view*.
> > > > - **Reply to W3** — To refute the reviewer’s unsubstantiated disagreement with a statement in our manuscript.
> > > > - **Reply to W4** — To reiterate the point of the five steps shown on p. 6 of the original manuscript (now on p. 6 – 7, ln. 306 – 325) to aid the reviewer’s understanding, with five references included.
> > > > - **Reply to W5** — To defend our flow of presentation of Alg. 1 and Alg. 2 while remaining open to the reviewer’s constructive suggestion(s) on how else to improve.
> > > > - **Reply to W6** — To correct the reviewer’s erroneous claim that *The framework predicts only the magnitude of the velocity in 2D*, pointing out that our framework predicts all primitive variables and the words “velocity magnitude” have never been used in any version of our manuscript.
> > > > - **Reply to Q1** — To answer that our datasets were generated with the RANS methodology, as clearly written in Appendix A, p. 13, ln. 675 – 676 of the original (and revised) manuscript, in contrary to the reviewer’s claim that *Did they use RANS or any other averaging method?! This has been totally missed in the manuscript.*
> > > > - **Reply to Q2** — To remind the reviewer on our work’s objective and the wide range of distance between the two airfoils that was covered in this work to counter the reviewer mistaken remark that *However, having two airfoils at a relatively large distance is not really a complicated geometric domain.*
> > > > - **Reply to Q3** — To answer the reviewer’s questions on the effects of reducing the distance between the two airfoils and increasing the angle-of-attack (AOA) with quantitative analyses of our results presented in the original manuscript that our framework remains accurate for in these scenarios.
> > > > - **Concerns in Q3** — To (i) point out that the reviewer’s questions on the effects due to airfoil thickness and AOA magnitude are different inquiries than Q3; (ii) provide quantitative answer that our framework is not sensitive to thickness and AOA, for which we have tested for a range that is neither *thin* (for thickness) nor *very small* (for AOA) by conventional standards (references provided); and (iii) refute the insinuation of an attempt to mislead our readers.
> > > > - **Other Scenarios** — To highlight that the additional cases suggested by the reviewer are out of scope and not possible to be evaluated due to a lack of data.
> > > > - Closing — To (i) repeat that our submission is a proof-of-concept for our novel method, so we should focus our work instead of applying our method over many different configurations; and (ii) request respectfully for the reviewer to provide constructive comments for us to improve our work, which has gone unheeded.

---

> > > > ### Author Response · Authors · 2024-12-02
> > > >
> > > > Next, the reviewer commented that:
> > > >
> > > > *They now also argue that they did not conduct the experiments because it would change their submission a lot, which is not true.*
> > > >
> > > > We have explicitly stated in the Abstract and Introduction (now on p. 1, ln. 17 and ln. 43 – 47 of the revised manuscript) that our scope is in the tandem-airfoil configuration, which is drastically different from the scenarios of Refs. [8, 9] suggested by the reviewer. Hence, conducting experiments on them will be the exact meaning of going out of scope. Once again, our submission is a feasibility study of our novel method and such works typically focus on one to two scenarios, as can be seen in published works of both AI and fluid communities [1 – 7, 10 – 11]. Furthermore, the reviewer's requests on the new experiments were not mentioned in the original comments, but in their response that was sent 6 hours before the deadline for revisions of the manuscript. However, we remain confident in the applicability of our framework, so much so that we have openly accepted the reviewer’s challenge to test our framework in the additional cases, as long as the reviewer can provide the relevant and appropriate training datasets.
> > > >
> > > > =====
> > > >
> > > > In conclusion, provided that the reviewer does not make any more insidious and, more importantly, ungrounded comments on our research integrity and contributions, we will be glad to end this discussion and leave the final decision to the Chair.
> > > >
> > > > References:
> > > > 1. W. Cao, J. Song, and W. Zhang, Solving high-dimensional parametric engineering problems for inviscid flow around airfoils based on physics-informed neural networks, J. Comput. Phys. 516 (2024) 113285
> > > > 2. X. Ren, P. Hu, H. Su, F. Zhang, and H. Yu, Physics-informed neural networks for transonic flow around a cylinder with high Reynolds number, Phys. Fluids 36 (2024) 036129
> > > > 3. X. Peng, X. Li, X. Chen, X. Chen, W. Yao, A hybrid deep learning framework for unsteady periodic flow field reconstruction based on frequency and residual learning, Aerosp. Sci. Technol. 141 (2023) 108539
> > > > 4. F. Bonnet, J. Mazari, P. Cinnella, P. Gallinari, AirfRANS: High fidelity computational fluid dynamics dataset for approximating Reynolds-averaged Navier–Stokes solutions, Adv. Neural Inf. Process. Syst. 35 (2022) 23463–23478
> > > > 5. J. Chen, E. Hachem, J. Viquerat, Graph neural networks for laminar flow prediction around random two-dimensional shapes, Phys. Fluids 33 (2021) 123607
> > > > 6. D. Kochkov, J. A. Smith, A. Alieva, Q. Wang, M. P. Brenner, S. Hoyer, Machine learning–accelerated computational fluid dynamics, PNAS 118 (2021) e2101784118
> > > > 7. O. Obiols-Sales, A. Vishnu, N. Malaya, A. Chandramowliswharan, CFDNet: A deep learning-based accelerator for fluid simulations, in: Proceedings of the 34th ACM International Conference on Supercomputing, ICS ’20, 2020
> > > > 8. S. Sarkar, N. Gupta, K. Debnath, P. R. L. Raj, Computational analysis of flow around two wall-mounted trapezoidal bluff bodies arranged in tandem position, J. Fluids Eng. 147 (2025) 021301
> > > > 9. S. Gupta, R. Naveen, B. V. Prathik, and S. Spurthy, CFD Simulation of turbulent flow around multi-element airfoil, AIP Conf. Proc. 2204 (2020) 03008
> > > > 10. R. Girshick, J. Donahue, T. Darrell, J. Malik. Rich feature hierarchies for accurate object detection and semantic segmentation, in: Proceedings of the IEEE conference on computer vision and pattern recognition, CVPR, 2014, 580–587
> > > > 11. I. Goodfellow, J. Pouget-Abadie, M. Mirza, B. Xu, D. Warde-Farley, S. Ozair, A. Courville, and Y. Bengio, Generative adversarial nets,  Adv. Neural Inf. Process. Syst. 27 (2014) 2672–2680

---

> ### Author Response · Authors · 2024-12-02
>
> We thank the reviewer for the reply and finalizing their decision. We too do not want to drag the discussion out longer than necessary, but there are several false points in the reviewer response that we see the need to clarify. We understand that our clarifications will not have any effect on the reviewer’s rating, and that is not our intent anyway.
>
> The reviewer said:
>
> *This is 2D, predicts flow around objects that have a very limited effect on each other.*
>
> Many recent published works [1 – 7] that focus on their methods are in 2D, and that did not negate their contributions. Our framework, even though still for 2D, is novel by proving that an NN can be trained to **provide a close estimate to the ground truth of a tandem airfoil case by reusing its corresponding pair of single airfoil results, which are (i) cheaper to generate and (ii) more widely available**. The value of our framework is supported by Reviewers kz1T and 1Tte, stating that:
>
> *The authors explore how to decompose the problem to enhance the current neural PDE method, which is a underexploited direction.* and
>
> *The method is targeted at a relevant engineering application, with tandem airfoils being common in aerospace and maritime engineering.*, respectively.
>
> We have tried to explain this key objective to the reviewer repeatedly, but to no avail.
>
> Regarding *limited effect on each other*, we have provided ample evidence with multiple references to refute the remark (see our previous response for **Concerns in Q3**). **The reviewer has just ignored all our points and continued with their own unsubstantiated opinion**. Hence, we will not engage in this meaningless exchange any further until there is/are quantitative proofs that the effect between the tandem airfoils is indeed limited.
>
> We believe that the necessity to our framework has already been captured in our **Reply to Q2**. By simply saying that *no one needs such a framework* is equivalent to disapproving the endorsements of our work by Reviewers kz1T and 1Tte:
>
> Comments of Reviewer kz1T
> - *The authors explore how to decompose the problem to enhance the current neural PDE method, which is a underexploited direction.*
> - *The method once again demonstrates the efficiency of gradually building results by incorporating prior information, such as freestream velocity fields or knowledge learned from simpler problems, like fields involving a single geometry.*
>
> Comments of Reviewer 1Tte
> - *The residual and domain decomposition techniques effectively improve performance consistently (table 3 and table 4).*
> - *The multi-NN inference slightly helps to increase accuracy (table 4)*
> - *The method definitely improves over the mesh graphnet baseline on the more challenging Random cruise setting (table 5).*
>
> Of course, the reviewer certainly entitled to their view that *no one needs such a framework*. We shall only repeat our caveats for the so-called *easiest way*:
> - Our results (see Tabs. 3 and 5 of the original and revised manuscripts) have demonstrated that, in the context of tandem-airfoil configuration, our framework performs much better than direct learning from multiple bodies (the reviewer’s suggestion).
> - Databases suitable for AI training of multiple bodies are scarce, and, as much as the reviewer disregards them, the four tandem-airfoil datasets are first of their kind to the best of our knowledge, and we are keen to share them once this submission is published.
>
> On the latter point, note that Reviewers kz1T and 1Tte have both confirmed the importance of our datasets, saying that:
>
> *The paper also introduces a new set of datasets, which would represent a valuable contribution to the community if made available as open-source.* and
>
> *I think the simulated datasets are also very valuable for the community. I will increase my score to 6.*, respectively.
>
> The reviewer then went on with a blatantly false claim:
>
> *Instead of taking my questions seriously at the beginning of the discussion period*.
>
> We always take the comments and questions of our reviewers seriously, regardless of the phase of submission, as can be seen in our replies to the reviewer’s questions, dated Nov 21, 2024 (between 14:48 to 15:12 AOE) and Nov 29 – 30, 2024, (between 20:43 to 03:59 AOE), which were submitted after multiple internal discussions and revisions. Unlike the reviewer’s habitually baseless responses, these replies comprehensively comprise of additional analyses, qualitative and quantitative results, and new references. All of our inputs are publicly accessible, so other researchers can objectively see our seriousness.

---

### Official Review · Reviewer_1Tte · 2024-11-01

**Soundness:** 2
**Presentation:** 2
**Contribution:** 2
**Rating:** 6
**Confidence:** 4

**Summary:**

The paper introduces a method to improve flow field prediction in complex geometries by leveraging a decomposition into simpler ones, specifically for tandem-airfoil configurations. The core idea of the framework is to train two neural networks, one to predict the flow field around individual airfoils, and the second one to predict the flow field around the tandem airfoil (i.e. the combination of the two). The authors adopt a two-step straining approach, where they first train the network on simple geometries and then train the network on the more complex ones. In both cases, the models are formulated with a residual approach, i.e. the output is expressed as $U= U_{est} + U$. In the first step, $U_{est} = U_{\infty}$ and in the second step $U_{est} = \tilde{U}$, where   $\tilde{U}$ is the smooth combination of the two predictions of the first model. The authors detail a multi-NN inference procedure to predict the flow field successively at the front, back, upper, lower regions before predicting the final predictions. This helps reduce memory needs. They train the models on 4 different datasets that encompass varying Reynolds and angle of attack (AoA) configurations.

**Strengths:**

* The method is targeted at a relevant engineering application, with tandem airfoils being common in aerospace and maritime engineering.
* The residual and domain decomposition techniques effectively improve performance consistently (table 3 and table 4).
* The multi-NN inference slightly helps to increase accuracy (table 4)
* The method definitely improves over the mesh graphnet baseline on the more challenging Random cruise setting (table 5).

**Weaknesses:**

* The approach combines existing techniques, such as SV and DID features, pre-training, residual learning, and domain decomposition. However, the paper reads more like a catalog of these techniques rather than a cohesive, novel framework, and the overall technical contribution is weak.

* The paper lacks a clear explanation of the smooth-combining method’s validity. For example, would we expect to get an interpolating weight such as the one described by Figure 2 or is it surprising ?

* The proposed techniques are computationally intensive at inference, particularly with 18 seconds required for geometric feature computation and 9 seconds for smooth-combining, limiting their practical applicability compared to standard solvers.

**Questions:**

* Could you compare the smooth-combining method to a simple linear interpolation weighted by the distance to each airfoil? What makes smooth-combining preferable, or why might it make more sense in this context?

* Why is it necessary to compute a single DID for both objects simultaneously? Would it be feasible to calculate a DID for each object independently and use these two as separate inputs instead of a combined one?

* How sensitive is the overall training procedure to the hyperparameters used at each stage? Some insights into this sensitivity could clarify stability and robustness.

* Given the multi-network setup with separate neural networks for each domain part, does this pipeline increase the risk of overfitting, particularly when applied to each sub-domain individually?

---

> ### Author Response · Authors · 2024-11-22
>
> **Reply to W1**:
> Thank you for the comment. The impression of a lack of a cohesive framework may have stemmed from our presentation style, which does not explicitly use terms like "framework" or "unified scheme". However, the proposed method is, in fact, a curriculum learning scheme, designed to decompose the training process of complex geometries into simpler steps. This enables reuse of CFD simulations of simple geometries and incorporation of physical priors, including freestream prior knowledge, to enhance the training process. As Reviewer kz1T has pointed out, "how to decompose the problem to enhance the current neural PDE method ... is a underexploited direction” and, to the best of our knowledge, no prior work has addressed this problem.
> Beyond the overall curriculum learning structure (Fig. 1 of the manuscript), our contribution includes exploiting freestream priors in the first stage of training to ease the learning process, which is a novel application in physics-informed AI aimed at accelerating simulations compared to traditional solvers that are computationally expensive for complex objects. Furthermore, in residual training, our smooth-combining method (Sec. 3.1 of the manuscript) introduces the use of freestream priors to integrate simulations from simpler geometries, another contribution that, to our knowledge, has not been explored in previous works. For directional integrated distance (DID), we developed Algorithm 2 to overcome the limitations of the original algorithm, which will struggle with numerical complexity when handling multiple objects. More precisely, the original algorithm requires 5222 seconds to compute DID of tandem airfoils. Without the advancement of Algorithm 2, DID would be unsuitable for our learning task. While our method seems to be an integration of  existing techniques like pre-training and residual learning, the curriculum learning structure and innovations, such as freestream priors and smooth-combining, represent significant and novel technical contributions that have not been exploited in previous works. In addition to technical contributions, for this study, we established four new tandem airfoil datasets with over 2000 CFD simulations. The datasets will be made available to the public once this work is published. We believe that they are valuable resources to the community, since very limited CFD datasets tailored for physics-informed AI studies are currently available. To address this concern, we have revised our paper to emphasize our novelties, contributions, and cohesive learning process.

---

> ### Author Response · Authors · 2024-11-22
>
> **Reply to W2 and Q1**:
> Thank you for the comment. To validate the effectiveness of the smooth-combining method, we compared its performance against freestream and a simple linear interpolation weighted by the distance to each airfoil as defined in Equation (3),
> \begin{equation}
> \begin{aligned}
>     \widetilde{\boldsymbol{U}}(i) &= \boldsymbol{\gamma}(i)\cdot \boldsymbol{U}_1(i) + \big(1-\boldsymbol{\gamma}(i)\big) \cdot \boldsymbol{U}_2(i),\\
>     \boldsymbol{\gamma}(i) &= \frac{d_2(i)}{d_1(i) + d_2(i)} , \\
> \end{aligned}
> \end{equation}
> where $d_1$ and $d_2$ are the shortest distances to front (leading) and back (trailing) airfoils, respectively. Figure 10 in Appendix H of the revised manuscript illustrates the weighting field, $\gamma$, generated from distance-based linear interpolation, showing a smooth gradient between the two airfoils.
>
> The comparison between (a) freestream, (b) distance-based linear interpolation, and (c) smooth-combining methods is presented in Fig. 11 in Appendix H of the revised manuscript, which shows the absolute error contours of the combined velocity components and pressure fields relative to their corresponding ground truths. The smooth-combining method demonstrates the lowest errors, particularly in the downstream and flow interaction regions, where the freestream and linear interpolation methods show pronounced inaccuracies.
>
> This qualitative observation is supported by the quantitative results in Tab. 14 in Appendix H of the revised manuscript, where the smooth-combining method achieves the lowest mean absolute error (MAE) against ground truth across all evaluated metrics, including velocity components and pressure. Specifically, the smooth-combining method outperforms both freestream and linear interpolation with an overall  MAE $(\times10^{-3})$ of $1.46\pm0.31$, compared to $1.76\pm0.23$ and $6.45\pm0.53$ for linear interpolation and freestream, respectively.
>
> To further assess the utility of smooth-combining, we conducted an additional experiment using the linear-interpolated flow fields as initial estimators for training the model (MGN + PRE-RES-FREE + RES-COMB) on the Cruise AOA=$5^\circ$ dataset. As shown in Tab. 15 in Appendix H of the revised manuscript, the smooth-combining approach results in significantly lower MSE compared to linear interpolation. These results confirm that the smooth-combining method not only provides a more accurate starting point for further training, but also captures complex flow interactions more effectively than the alternative approaches. Its superior performance in both initial approximation and subsequent training highlights its importance for this framework.
>
> Table 14:
>  Methods / MAE | $u(\times10^{-2})$ | $v(\times10^{-3})$ | $p(\times10^{-4})$ | Overall $(\times10^{-3})$
> :----------------------:|:--------------------:|:--------------------:|:--------------------:|:-------------------------:
>  Freestream             | $1.57\pm0.13$        | $3.08\pm0.28$        | $5.45\pm0.58$        | $6.45\pm0.53$
>  Linear Interpolation   | $0.39\pm0.05$        | $1.14\pm0.15$        | $1.95\pm0.25$        | $1.76\pm0.23$
>  Smooth-Combining       | $0.31\pm0.07$        | $1.07\pm0.16$        | $1.71\pm0.31$        | $1.46\pm0.31 $
>
> Table 15:
>  Methods / Datasets   | Cruise AOA=$5^\circ$
> :--------------------:|:--------------------:
>  Linear Interpolation | $1.15\pm0.58$
>  Smooth-Combining     | $0.67\pm0.39$
>
> The above discussion has been added as Appendix H in the revised manuscript to provide better clarity to our work.

---

> ### Author Response · Authors · 2024-11-22
>
> **Reply to W3**:
> Thank you for the comment regarding the computational cost of our method. While it is true that geometric feature computation and smooth-combining steps currently require 18 seconds and 9 seconds, respectively, these operations—such as (i) reading CFD files, (ii) calculating geometric features, and (iii) smooth-combining fields—are performed on CPU and have not been optimized. We anticipate further reductions in computational time with future optimizations or through the use of hardware accelerations, such as GPUs.
>
> It is important to note that, in general, real-time simulation is not a requirement, since CFD simulation is extensively used for component designs, e.g., in the aerospace industry. However, the traditional simulators are extremely computationally costly, which prevents simulating complex components. In our experiments, as shown in Tabs. 8 and 9 of the revised manuscript, the proposed method reduces 76% of simulation time compared to OpenFOAM run in parallel with 64 cores. Although the baseline networks, MGN and IVE, can provide higher simulation speed, their performance is unacceptable (see Tab. 3 of the revised manuscript). More clearly, their average MSEs are 499% and 280% respectively higher than the proposed method. Even pretraining on single airfoils, their errors are 199% and 256% respectively higher than the proposed method.
>
> The significant improvements on speed over the traditional simulator and on accuracy over other the baselines highlight the practical applicability of our approach for scenarios requiring multiple simulations such as design optimization or control.
>
> **Reply to Q2**:
> Thank you for the insightful question. Computing a single DID for both objects simultaneously is primarily a practical decision aimed at improving efficiency and scalability. While Algorithm 1 can technically compute a single DID for multiple objects simultaneously, its numerical complexity increases significantly with each additional object, resulting in slower training and inference speeds. For instance, in our experiments, Algorithm 1 required 5222 seconds to compute the DID for tandem airfoils, whereas Algorithm 2, which computes separate DIDs for individual airfoils and then combines them, completed the task in only 3.5 seconds. Additionally, calculating separate DIDs for each object would increase the input size proportionally to the number of objects. If the dimension of a single object’s DID is $N$ and there are $M$ objects, the total input size would scale as $N\times M$, leading to higher memory requirements and computational load on GPUs. By combining the DIDs into a single representation, our approach maintains scalability and significantly reduces computational overhead. Algorithm 2 strikes an effective balance, allowing for efficient handling of multi-object scenarios without sacrificing performance as discussed in Appendix E of the original (and revised) manuscript.
>
> To further evaluate the feasibility of using individual DIDs for each object, we conducted an experiment comparing the performance and resource usage of individual DIDs versus a single combined DID using the model MGN+PRE-RES-FREE+RES-COMB on the Cruise AOA=$5^\circ$ dataset. The results, tabulated below, reveal that the single combined DID achieves better computational efficiency and prediction accuracy. These results have been provided as Appendix G in the revised manuscript.
>
> Table 13:
> Methods    | Average GPU memory usage (GB) | MSE $(\times 10^{-2})$
> :-------------------:|:-----------------------------:|:----------------------:
>  Single combined DID | $16.64$                         | $0.67\pm0.39$
>  Individual DID      | $23.37$                         | $0.80\pm0.42$

---

> ### Author Response · Authors · 2024-11-22
>
> **Reply to Q3**:
> Thank you for the question. To ensure a fair comparison with the baseline methods, hyperparameters such as learning rate, network depth, and layer sizes were kept consistent with the baseline settings. This minimizes variability and ensures that the observed improvements are due to our proposed approach rather than hyperparameter tuning.
> However, sensitivity to certain domain-specific parameters, such as the maximum DID distance ($d_{max}$) and the number of angle segments used in DID computation, could impact performance. These parameters influence the granularity of the directional distance representation and its ability to capture relevant physical interactions.
> Sensitivity studies were conducted using the model MGN+PRE-RES-FREE+RES-COMB on the Cruise AOA=$5^\circ$ dataset. The results, shown in Tab. 12 in Appendix F of the revised manuscript, reveal the impact of varying $d_{max}$ and the number of angle segments on MSE.
>
> Table 12:
>  $d_{max}$ | No. of angle segments | MSE$(\times10^{-2})$
> :---------:|:---------------------:|:--------------------:
>  $5$         | $8$                     | $0.67\pm0.39$
>  $2.5$       | $8$                     | $0.85\pm0.38$
>  $5$         | $4$                     | $0.89\pm0.44$
>  $5$         | $16$                    | $0.51\pm0.29$
>
> Reducing $d_{max}$ to 2.5 increases MSE, likely because a smaller $d_{max}$ limits the model’s ability to capture longer-range interactions. Similarly, decreasing the number of angle segments to 4 also leads to higher errors, suggesting that fewer angle segments reduce the directional resolution of the DID representation. In contrast, increasing the number of angle segments to 16 improves the performance at the expense of higher computation time for the DID features. Compared with the baseline MGN, whose MSE is $2.08\times10^{-2}$ on the same Cruise AOA$=5^\circ$ dataset, the variations of these results are relatively minor, indicating the proposed method's robustness and insensitivity to these parameters.
>
> **Reply to Q4**:
> Thank you for raising the concern about the risk of overfitting in our multi-network setup. Based on our experiments, overfitting does not appear to be an issue. As shown in Fig. 6 in Appendix D of the revised manuscript, the training and validation loss curves for each sub-domain (front, back, upper, and lower) exhibit consistent convergence. For the front and back sub-domains (Figs. 6a and 6b), the validation loss closely follows the training loss across 200 epochs, indicating good generalization to unseen data. Similarly, for the upper and lower sub-domains (Figs. 6c and 6d), both losses converge rapidly and remain stable, suggesting no signs of overfitting. It is worth noting that the size of the computational domain (graph) is large. Prior to domain decomposition, the model may have been underfitting, and this structural decomposition allows the networks to effectively capture sub-domain-specific flow features without increasing the risk of overfitting. These results collectively demonstrate that our approach maintains a balance between model complexity and generalization.

---

> > ### Comment · Reviewer_1Tte · 2024-11-25
> > **Response**
> >
> > I appreciate the authors' response. Though the method section could be improved for improving the clarity of the manuscript, the additional results have addressed most of my concerns. I think the simulated datasets are also very valuable for the community.
> > I will increase my score to 6.

---

> > > ### Author Response · Authors · 2024-11-25
> > >
> > > We sincerely thank you for your thoughtful feedback and the upgraded score. Your insightful suggestions have been instrumental in improving the clarity and quality of our manuscript. We will carefully integrate your valuable advice into the finalized manuscript. Thank you for your support and recognition of our work!

---

### Official Review · Reviewer_kz1T · 2024-11-01

**Soundness:** 3
**Presentation:** 2
**Contribution:** 3
**Rating:** 6
**Confidence:** 4

**Summary:**

The paper introduces a neural PDE approach for predicting CFD velocity fields around complex geometries using domain decomposition and blending techniques. The neural network is first pre-trained to predict the velocity field for a single geometry, and these predictions are then combined using a weighted approach for different velocity fields. Additionally, the method explores techniques to aid training, such as incorporating freestream velocity fields or leveraging pre-trained fields.

**Strengths:**

- The authors explore how to decompose the problem to enhance the current neural PDE method, which is a underexploited direction.
- The paper also introduces a new set of datasets, which would represent a valuable contribution to the community if made available as open-source.
- The method and process are presented with sufficient clarity to allow for a good understanding of the model.
- The method once again demonstrates the efficiency of gradually building results by incorporating prior information, such as freestream velocity fields or knowledge learned from simpler problems, like fields involving a single geometry.

**Weaknesses:**

- The multi-NN approach is applied with a highly specific dependency, as illustrated in Fig. 3 (p. 6). It would be helpful to explain the rationale behind this dependency. Additionally, it appears that this dependency is closely tied to the geometric setting introduced in the paper. The authors should either discuss this as a limitation or specify the prior information required to determine this dependency.
- Please clarify how the features, including prior information (e.g., inlet velocity, DID) and previous predictions (e.g., front/back prediction), are input into the network.
- Recently, some works have addressed the challenges of complex geometry and domain decomposition, such as Mao et al. (2024). I suggest that the authors include a discussion of these studies.

References:
- Mao et al. (2024). Towards General Neural Surrogate Solvers with Specialized Neural Accelerators https://arxiv.org/abs/2405.02351

**Questions:**

See Weaknesses for questions.

---

> ### Author Response · Authors · 2024-11-22
>
> **Reply to W1**:
> The dependency structure of our multi-NN approach is designed specifically to address the inherent challenges in complex flow field predictions. By subdividing the domain into distinct front, back, upper, and lower flow fields, we leverage the local characteristics of each sub-domain to improve prediction accuracy and computational efficiency. The dependencies among these sub-domains are not arbitrary but rather carefully engineered to align with the directionality of the physical flow and the inlet boundary conditions. This structure enables us to sequentially propagate information through each sub-domain, ensuring that local predictions can build on previously predicted values in overlapping regions, which improves coherence across the overall flow field.
> As the reviewer pointed out, this dependency structure is tied to the specific geometric setting discussed in the paper, where the division of flow fields aligns with predefined boundaries and inlets. We recognize that, in more complex geometries or flow scenarios where different or dynamic boundary conditions might exist, this dependency structure may need adaptation. To apply this approach to arbitrary or complex geometries, prior knowledge of inlet positions, overlap regions, and flow directions is required. However, practitioners in this domain typically have encapsulated the necessary knowledge ensuring that this is rarely a significant limitation in practice.
> In response to this, we added a discussion in Sec. 3.4, p. 7, ln. 327 – 340 (highlighted in blue) of the revised manuscript acknowledging this as a limitation and specifying the prior information required for implementing this multi-NN structure for different geometries. This will guide future applications and adaptations of the multi-NN approach to broader CFD scenarios, where flexibility in domain subdivision and dependency configurations might be necessary.
>
> **Reply to W2**:
> We have discussed the detail of the input and neural network parameters in Appendix B. Repeating here for ease of discussion, each neural network model is dedicated to predicting a specific sub-region (front, back, upper, lower) of the CFD domain. To capture the localized flow dynamics and interactions across these sub-domains, we incorporate various input features, including geometry representations, flow properties, and overlapping predictions from neighboring regions. The input features (and their sizes) of all models consist of:
> 1. Node positions (2 features): Spatial location of each node, ensuring spatial coherence in predictions.
> 2. SV (2 features): Shortest vector from the solid boundaries to the node.
> 3. DID (8 features): Computed based on eight angle segments, each representing an arc of π/4 degrees. This discretization helps capture directional information at multiple orientations.
> 4. Freestream/combined fields (3 features): x-velocity, y-velocity, and pressure fields when residual training is applied to initialize predictions close to known physical priors.
> 5. Inlet/overlap velocity and pressure values (3 features): Inlet values are only provided to nodes located along the inlet boundary, while other nodes receive zero-valued inputs in this field. For overlapping regions, predicted values from adjacent flow fields are used as inputs, allowing each sub-domain NN to leverage the most recent predictions from neighboring regions.
>
> In total, making the input layer size of 15 without residual training and 18 with. Likewise, the number of output features of all models was 3, for the x-velocity, y-velocity and pressure fields.
>
> **Propagation of previous predictions**
> As noted, each sub-domain prediction serves as prior information for adjacent regions. Specifically, the Front NN receives inlet values as input features for nodes along the inlet, with all other nodes assigned zero values. The front prediction is then used as an overlap input feature for the Back NN, ensuring that the flow dynamics captured at the front are accurately propagated to the back region.
>
> In short, the inclusion of prior and predicted values in the input features enhances the model’s ability to capture dependencies and interactions between flow fields. For example, by using the front prediction in the back flow field, the model is better equipped to recognize continuity in the flow across sub-domains.

---

> ### Author Response · Authors · 2024-11-22
>
> **Reply to W3**:
> Recent works, such as Mao et al. (2024), have addressed challenges in solving large-scale PDE problems with complex geometries by employing domain decomposition techniques combined with neural operators. Their SNAP-DDM framework introduces a novel approach that utilizes overlapping subdomains with fixed grids and CNN-based architectures to generalize across diverse grid-based problem settings. While their method shares conceptual similarities with our use of domain decomposition, the strategies differ significantly in focus and application.
> Mao et al.'s work is designed for grid-based simulations, providing a broadly applicable framework with uniform subdomain division. In contrast, our multi-NN procedure is specifically tailored for mesh-based CFD simulations of tandem airfoils, employing case-specific domain decomposition optimized for performance in this specialized context. Unlike SNAP-DDM, which processes image-type input and output, our method handles mesh-based data, where flow fields cannot be treated as images. This distinction necessitates the use of different neural network architectures to effectively model the underlying physical phenomena.
> Another key difference is the emphasis of our work on reusing flow fields from simpler geometries as part of a curriculum learning-like framework (Fig. 1 of the manuscript). This approach leverages simple geometry flow fields to improve predictions for more complex geometries, a strategy not explored in SNAP-DDM. While SNAP-DDM focuses on broad generalizability using iterative subdomain solvers, our method prioritizes domain-specific optimizations for CFD applications, enabling efficient simulations in specialized cases like tandem airfoils.
> By tailoring our domain decomposition strategy and leveraging prior knowledge, our approach achieves performance improvements and computational efficiency in mesh-based CFD scenarios. In this revised paper, we have discussed Mao et al. (2024) in Sec. 2.2 “Related Work”.

---

> ### Author Response · Authors · 2024-11-27
> **Gentle Reminder**
>
> Dear reviewer kz1T,
>
> We sincerely appreciate the time and effort you have dedicated to reviewing our paper.
>
> As the discussion period approaches its end, we wanted to gently remind you that there are seven days remaining to share any additional comments or questions you might have. Your insights are invaluable, and we would be grateful for the opportunity to address any further concerns or feedback you may have before the discussion phase concludes.
>
> Thank you once again for your valuable contribution to this process.
>
> Best regards,
> Authors

---

### Meta-Review · Area_Chair_SYUj · 2024-12-23

**Metareview:**

The paper introduces a method for predicting flow fields around complex geometries, specifically tandem airfoils, by leveraging simulations from single-airfoil flow fields. The core idea is to train neural networks (NNs) to predict flow fields for single airfoil geometries and then combine these predictions to model the tandem configuration. The model employs a residual approach: for single airfoils, the NNs predict the residual with respect to a freestream condition (the undisturbed flow of a fluid in the absence of solid bodies). For tandem airfoils, the residual complements the combined prediction of the two airfoils. The authors claim these two ideas are novel. The model inputs include a geometric representation known as “Directional Integrated Distance” (DID), which has been previously used for single objects and is adapted here for multi-object scenarios. These ideas are implemented within a multi-NN framework. The computational domain is divided into predefined sub-fields, with each sub-field predicted by a separate neural network. The predictions from these sub-fields are then combined to produce the complete solution. Experiments compare the proposed approach using two NN backbones across four datasets, encompassing various airfoil shapes and dynamic configurations. Several ablation studies evaluate the importance of different components. The datasets used in the experiments will be released.

The reviewers acknowledge that this is a relevant engineering application, rarely explored in the recent literature on neural network-based partial differential equation (PDE) simulations, and that the method demonstrates some novelty, though it is somewhat incremental. The evaluations, particularly the ablation studies, highlight the relevance of the design choices. After carefully reviewing the manuscript, I agree that it demonstrates the effectiveness of the proposed ideas for tandem airfoils in simple yet diverse simulations. However, I have three main concerns: (i) technical presentation: The exposition could be improved for greater precision and to better target a broader machine learning (ML) audience; (ii) generality: While the framework appears principled, the configuration is highly specific, and there is no evidence it can be extended to more general problems; (iii) novelty: The technical contributions, while valuable, are relatively incremental.
Overall, while the paper has merits, in its current form, it is likely to appeal to a niche ML audience. I believe it would be better suited for resubmission after addressing these concerns.

**Additional Comments On Reviewer Discussion:**

The reviewers consider this a relevant but highly specific contribution with a somewhat incremental ML technical impact. Rz1T did not respond during the rebuttal. R1Tte acknowledged that the responses addressed several of their concerns but noted that the technical description could still be improved. I disregarded the review by R9Pbr, as it was both irrelevant and disrespectful. Subsequently, I carefully reviewed the manuscript, weighed the arguments presented by the reviewers, and considered my own assessment to reach a decision.

---

### Decision · Program_Chairs · 2025-01-22

Reject